# Quantum ground-state cooling of two librational modes of a nanorotor

Stephan Troyer [1], Florian Fechtel [1], Lorenz Hummer[1], Henning Rudolph[2], Benjamin A. Stickler [3]✉, Uroš Delić [1,4] & Markus Arndt [1]✉

Controlling the motion of nanoscale objects at the quantum limit promises opportunities to test fundamental quantum physics and advances in quantum sensing. Rotational motion is of particular interest, as its nonlinear dynamics in a compact, closed configuration space provides access to phenomena such as rotational interferometry, tunnelling between angular configurations and quantum-enhanced torque sensing. A key requirement for such experiments is the capability to trap nanorotors and cool their orientation close to the two-dimensional librational quantum ground state. When rotational motion is confined in a harmonic potential, it becomes librational. Here we demonstrate that coherent scattering into a high-finesse cavity enables the ground-state cooling of two orthogonal librational modes of an optically levitated $SiO_2$ nanoparticle. Using a laser-induced desorption loading technique, we trap and cool several dimers and trimers of silica nanospheres to their respective ground states, all within a single day. The simultaneous cooling of both librational degrees of freedom allows us to align an individual nanorotor with respect to a space-fixed axis with an angular precision better than 20 µrad—close to the quantum-mechanical zero-point fluctuations.

The quantum harmonic oscillator is among the most fundamental systems of physics and has been experimentally realized on a variety of mechanical platforms such as cantilevers[1] and bulk acousto-optic resonators[2]. Optically levitated nanoparticles provide a realization particularly close to the ideal harmonic oscillator. When trapped by Gaussian laser beams in high vacuum, they can achieve exceptionally high mechanical quality factors[3]. Their motion has been cooled to the quantum ground state in one[4–7] and two linear degrees of freedom[8,9] and recently also around one rotation axis[10].

For more than a decade, angular degrees of freedom have attracted increasing interest[11–13] because rotational dynamics introduces distinctive features to optomechanics. When driven to gigahertz rotation rates[14,15], nanorotors can probe material stress limits and enable ultra-sensitive torque measurements[16], paving the way for tests of vacuum friction[17,18], nanoscale magnetism and the search for non-Newtonian forces near surfaces[19]. In addition, a rapidly spinning nanorod has been used as a nanomechanical clock hand, acting as a local pressure sensor with micrometre spatial resolution[20].

As rotational motion follows nonlinear dynamics in a periodic, compact phase space, it enables distinct mesoscopic quantum phenomena[21], such as tunnelling in persistent quantum tennis racket flips[22], coherent coupling between spins and mechanical angular momentum[23–26], or rotational matter–wave interference, where the rotational wave functions of an initially aligned nanorotor can divide, expand and rephase without the need for beamsplitters or mirrors[27,28]. This opens alternative pathways for the preparation of massive Schrödinger cat states, which would be sensitive to models of wave-function collapse[29] or dark matter[30–32].

[1]University of Vienna, Faculty of Physics & Vienna Doctoral School in Physics & Vienna Center for Quantum Science and Technology, Vienna, Austria. [2]University of Duisburg-Essen, Faculty of Physics, Duisburg, Germany. [3]Institute for Complex Quantum Systems and Center for Integrated Quantum Science and Technology, Ulm University, Ulm, Germany. [4]Vienna Center for Quantum Science and Technology, Atominstitut, TU Wien, Vienna, Austria. ✉e-mail: benjamin.stickler@uni-ulm.de; markus.arndt@univie.ac.at

Although rotational quantum revivals have already been studied in molecular systems[33,34], observing similar quantum effects with more massive objects requires trapping, cooling and initializing them with alignment uncertainties close to the quantum zero-point fluctuation. This can be achieved by confining rotational motion in a two-dimensional (2D) harmonic potential, giving rise to librational oscillations through the interaction between an optical tweezer and the anisotropic polarizability of the particle. It has been proposed that such librational motion can be cooled by the coherent scattering of light into a high-finesse cavity[35,36]. Recent experiments have demonstrated the cooling of a $SiO_2$ particle to millikelvin temperatures in one[37] as well as of up to three[38–41] librational degrees of freedom, culminating in the coherent scattering cooling to a high-purity quantum ground state for a single librational mode[10].

Here we demonstrate the cooling of two librational degrees of freedom individually and show that both modes can be cooled simultaneously, such that the nanorotor's alignment is defined close to its zero-point uncertainty, a necessary condition for future experiments on rotational quantum interference and quantum-enhanced torque sensing.

Extending ground-state cooling from one to two librational modes requires implementing several key experimental advances. First, it is necessary to avoid hybridization between the two librational oscillation modes. We achieve this by coupling them to two orthogonal modes of a high-finesse optical cavity[40,42], which also enables the unambiguous identification of individual mechanical modes. This mechanism is specific to librational motion and does not exist for translational degrees of freedom, which necessarily couple to the same optical mode[8]. Precise control of the cavity birefringence allows the cavity modes to be aligned along predefined laboratory axes as well as tuning the birefringence-induced frequency splitting between them.

Second, although a high-finesse cavity enables efficient cooling, it also enhances the impact of laser phase noise, leading to excessive optical heating of mechanical motion. To overcome this limitation, we extend feedback-based phase-noise reduction schemes[10,43] to a multifrequency implementation, which is crucial for achieving 2D ground-state cooling.

Finally, we implement an improved laser-induced nanoparticle loading mechanism. It is similar to laser-induced acoustic desorption, which has been used in physical chemistry[44] and optomechanics before[45–47]. However, by reducing the thickness of the desorption layer from tens of micrometres to tens of nanometres, we lower the required laser pulse energy by about two orders of magnitude. This makes the method cleaner and more suitable for vacuum environments. As a result, the total experimental cycle time—from nanorotor launch and characterization in prevacuum to ground-state cooling in high vacuum—is reduced to less than an hour. This capability enables the ground-state cooling of several different nanorotors, including dimers, trimers and clusters of silica nanospheres, on the same day.

## Experimental setup

Our experimental platform is shown in Fig. 1a and discussed in more detail in the Methods and Extended Data Fig. 1. The nanorotors are assembled from two or more silica spheres with a nominal mean diameter of $d = 119 \pm 4$ nm (specified by microparticles GmbH) for cooling a single librational mode (one dimensional (1D)) and $156 \pm 5$ nm for cooling two librational modes (2D). Single spheres, dumbbells, trimers or clusters are launched at low pressure (~6 mbar) using laser-induced desorption (see the 'Robust and repeatable trap loading and cooling' section) and trapped in the optical tweezer light (wavelength $\lambda = 1,550$ nm) that propagates along the $z$ axis, linearly polarized along the $x$ axis.

We describe each nanorotor as an asymmetric rigid body with distinct moments of inertia and susceptibilities $\chi_a < \chi_b < \chi_c$. This accounts for imperfections in individual spheres[42], for birefringence[48] and for limitations of the Rayleigh–Gans approximation[35]. The orientation of

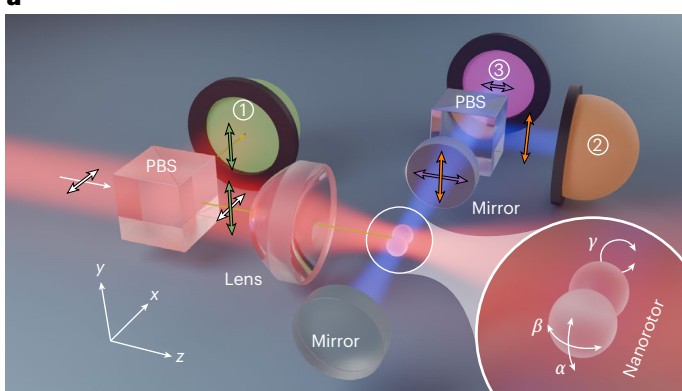

**a**

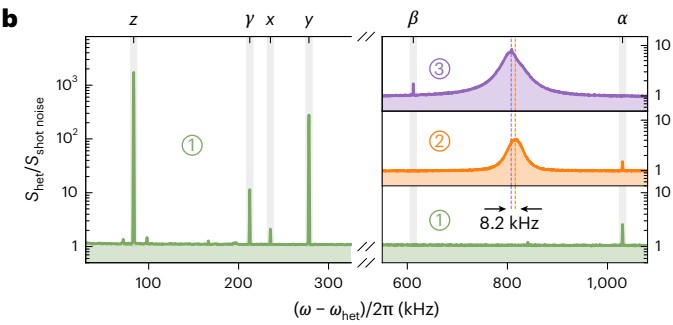

**b**

**Fig. 1 | Nanorotor trapped in an optical tweezer. a**, A silica nanorotor is trapped in an optical tweezer formed by light propagating along the $z$ axis and polarized along the $x$ axis, and focused using a high-numerical-aperture lens. The $y$-polarized backward-scattered light is collected and monitored on heterodyne detector 1. The coherently scattered tweezer light of the nanorotor populates an optical cavity, which is formed by two mirrors and oriented along the $x$ direction. The orthogonally polarized cavity modes are split using a PBS and monitored in heterodyne detectors 2 and 3. Inset: librational modes $\alpha$, $\beta$ and $\gamma$ in the defined reference frame. **b**, PSD $S_{het}$ of heterodyne detection of the backward-scattered light (1, green) and cavity modes polarized along the $y$ (2, orange) and $z$ (3, violet) axes show all the degrees of freedom. The spectra are taken at a tweezer–cavity detuning of $\Delta/2\pi \approx 800$ kHz, normalized to the shot-noise level $S_{shot\,noise}$ and shown here with respect to the heterodyne frequency $\omega_{het}$. Librations $\alpha$ and $\beta$ are visible in different cavity modes, which are frequency separated by ~8.2 kHz. Note that the birefringence splitting aligns the cavity modes such that the higher-frequency cavity mode couples to the higher-frequency librational $\alpha$ mode and the lower-frequency cavity mode to the lower-frequency $\beta$ mode.

the particle in the space-fixed laboratory frame ($\mathbf{e}_x, \mathbf{e}_y, \mathbf{e}_z$) is specified by the three Euler angles ($\alpha, \beta, \gamma$) in the $z$–$y'$–$z''$ convention (Fig. 1a, inset). The nanorotor aligns its figure axis along the linear polarization of the tweezer, which strongly traps the libration along $\alpha$ and $\beta$ at frequencies in the range of 0.5–1.3 MHz (Fig. 1b).

The nanorotor is trapped in the centre of an optical cavity that is oriented along the $x$ axis. It has a finesse of $\mathcal{F} \approx 300,000$, an energy decay rate of $\kappa/2\pi = 32.4$ kHz and a detuning of $\Delta = \omega_c - \omega_l$ with respect to the tweezer frequency $\omega_l = 2\pi c/\lambda$ (Fig. 1a). The cavity is slightly birefringent and its eigenmodes are polarized along the $y$ and $z$ axes with a frequency splitting of about 8.2 kHz. The nanorotor scatters the tweezer light into an initially empty cavity mode, thereby coupling its motion to the cavity field via photon recoil. The interaction between the particle and cavity is described by the coupling constants $g_\alpha \propto \chi_c - \chi_a$ and $g_\beta \propto \chi_c - \chi_b$ (equation (9)) and the interaction potential

$$\frac{U_{int}}{\hbar} = (g_\alpha a_y + g_\alpha^* a_y^\dagger)(b_\alpha + b_\alpha^\dagger) + (g_\beta a_z + g_\beta^* a_z^\dagger)(b_\beta + b_\beta^\dagger), \quad (1)$$

where $b_\mu^\dagger$ and $a_\nu^\dagger$, with $\mu \in \{\alpha, \beta\}$ and $\nu \in \{y, z\}$, are the creation operators of the harmonic oscillators that describe the librational motion in the

optical trap and the photon field in the cavity, respectively (Methods). The interaction Hamiltonian in equation (1) reveals that the two librational degrees of freedom couple selectively to orthogonal cavity polarizations: $\alpha$ libration to the $y$-polarized mode $a_y$ and $\beta$ libration to the $z$-polarized mode $a_z$. All translational modes couple to the same shared cavity mode $a_y$ (ref. 8).

The total Hamiltonian describes the energy associated with the population of the two cavity modes, the two mechanical modes of frequency $\Omega_\mu$, as well as the interaction between them:

$$H = \sum_{\nu=y,z} \hbar\Delta a_\nu^\dagger a_\nu + \sum_{\mu=\alpha,\beta} \hbar\Omega_\mu b_\mu^\dagger b_\mu + U_{int}. \qquad (2)$$

To unambiguously identify the two librational modes $\alpha$ and $\beta$, we split the transmitted cavity light via a polarizing beamsplitter (PBS) into the cavity's eigenmodes $a_y$ (detector 2) and $a_z$ (detector 3). Each signal is mixed with a local oscillator (LO) detuned by $\omega_{het}$ to yield the heterodyne signal displayed in Fig. 1b (Methods and Extended Data Fig. 1c). On the basis of the theoretical prediction, we identify the peak appearing in the orange (violet) power spectral density (PSD) of $a_y$ ($a_z$) as $\alpha$ ($\beta$). For the trapped $SiO_2$ cluster shown in Fig. 1b, we clearly identify the frequencies $\Omega_\beta = 2\pi \times 612$ kHz and $\Omega_\alpha = 2\pi \times 1{,}030$ kHz. Since the cavity transmission is affected by cavity-enhanced laser phase noise (Methods and Extended Data Fig. 2c), we use the heterodyne detection of the backward-scattered tweezer light (detector 1) to monitor all degrees of freedom during the rest of the experiment.

When the cavity is blue-detuned relative to the tweezer light ($\Delta > 0$), the interaction Hamiltonian in equation (2) predicts cooling via coherent scattering[49–51]. At resonance, the interaction rate between the particle and cavity light field increases. If the detuning to the tweezer matches the level spacing of the mechanical harmonic oscillator, the probability of anti-Stokes scattering is maximized. Each anti-Stokes scattering event reduces the oscillator quantum number by one, $n \to n-1$ and increasing the photon energy from $\hbar\omega_l$ to $\hbar(\omega_l + \Omega_\mu)$. Starting from a thermal state, the net effect of these individual transitions is a decrease in the mean phonon occupation $\langle n \rangle$, that is, cooling of the mechanical oscillator, in close analogy to early experiments with optical cooling in other nanomechanical systems[52]. By contrast, Stokes scattering raises the oscillator quantum number and reduces the photon energy, thereby heating the nanorotor motion. However, such photons are off-resonant with respect to the cavity and, therefore, suppressed. On average, the resulting imbalance transfers mechanical energy to the optical field, raising the energy of the photons that leave the cavity[49,53].

The minimum phonon occupation $n_\mu$ is reached when the cooling rate balances the heating rate. The net cooling rate is determined by the difference between the Stokes $A_\mu^+$ and anti-Stokes $A_\mu^-$ scattering rates. Heating arises from collisions with background gas and photon recoil, which together contribute $\Gamma_\mu$, as well as Stokes scattering. We reduce the influence of gas collisions by pumping the chamber to about $3 \times 10^{-8}$ mbar. The equilibrium phonon occupation of the librational modes $\mu \in \{\alpha, \beta\}$, including a laser phase-noise contribution $n_\phi(\Omega_\mu)$ (ref. 54), is then given by

$$n_\mu = \frac{\Gamma_\mu + A_\mu^+}{A_\mu^- - A_\mu^+} + n_\phi(\Omega_\mu). \qquad (3)$$

The phase-noise contribution is approximately $n_\phi(\Omega_\mu) \approx S_\phi(\Omega_\mu)n_{cav}/\kappa$ and determined by the spectral density of the laser phase noise $S_\phi(\Omega_\mu)$ and the intracavity photon number $n_{cav}$ (refs. 54,55). To reduce the effect of phase noise, we actively stabilize the laser phase using feedback derived from an unbalanced Mach–Zehnder interferometer[10,43] (Extended Data Fig. 1b). In this way, we are able to reduce the noise background by more than two orders of magnitude at the eigenfrequencies of both $\alpha$ and $\beta$ oscillations (Methods and Extended Data Fig. 2b,c).

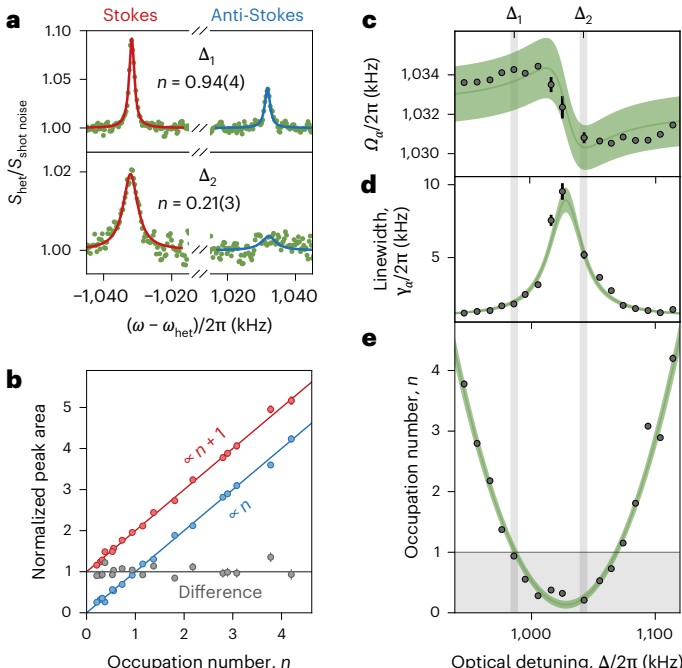

**Fig. 2 | 1D ground-state cooling. a**, PSD of the Stokes (red) and anti-Stokes (blue) scattering process for $\Delta_1/2\pi = 986$ kHz and $\Delta_2/2\pi = 1{,}042$ kHz. The imbalance of the peak heights is used to extract the occupation $n$. **b**, Stokes (red) and anti-Stokes (blue) scattering rates for different occupations $n$ show their proportionality to $n+1$ and $n$. **c–e**, Fitted mechanical frequency (**c**), linewidth (**d**) and obtained occupation (black points; **e**) are shown as a function of the tweezer–cavity detuning $\Delta$. Error bars in **b**–**e** denote the $1\sigma$ uncertainties from individual fits and are smaller than the marker size for most data points. Green regions denote the $1\sigma$ uncertainty of the global fit, which accounts for uncertainties in linewidth, coupling, offset frequency, heating rate and phase-noise occupation. The grey-shaded region marks a ground-state population with a probability greater than 50%.

## 1D ground-state cooling

For the $SiO_2$ cluster shown in Fig. 1b, we record the cooling of $\alpha$ libration as a function of detuning $\Delta$ around $\Omega_\alpha$. In Fig. 2a, we plot the PSD of this motion at positive and negative frequencies to perform sideband thermometry with the Stokes (red) and anti-Stokes (blue) sidebands. We fit the peaks to a Lorentzian profile to extract their frequencies, linewidths and background noise levels.

The Stokes and anti-Stokes scattering rates, which are proportional to the respective peak areas, scale with the occupation number $n$ as $A^+ \propto n+1$ and $A^- \propto n$. The occupation number $n$ can, therefore, be extracted from the sideband amplitudes (Methods).

Already at a cavity detuning of $\Delta_1/2\pi = 986$ kHz, the imbalance between the red and blue sidebands is substantial, signalling a phonon occupation of $n = 0.94 \pm 0.04$. Shifting the detuning to $\Delta_2/2\pi = 1{,}042$ kHz leads to an occupation of $\alpha$ libration as low as $n = 0.21 \pm 0.03$, which corresponds to a probability of populating the harmonic oscillator's ground state of $83 \pm 2$ %.

Figure 2b shows the normalized peak areas as a function of the occupation number. As expected for a quantum harmonic oscillator, the probabilities for Stokes and anti-Stokes scattering scale with the occupation as $n+1$ and $n$, respectively. The stability and repeatability of the experiment are evidenced by plotting the difference (grey) of the normalized sidebands for different cavity detunings and, hence, occupation numbers. The difference remains consistently close to unity.

Scanning the cavity detuning around mechanical resonance changes the optomechanical coupling and, therefore, both eigenfrequency (Fig. 2c) and linewidth (Fig. 2d) of librational motion. Forces due to the intracavity light field act like a frequency-tunable optical

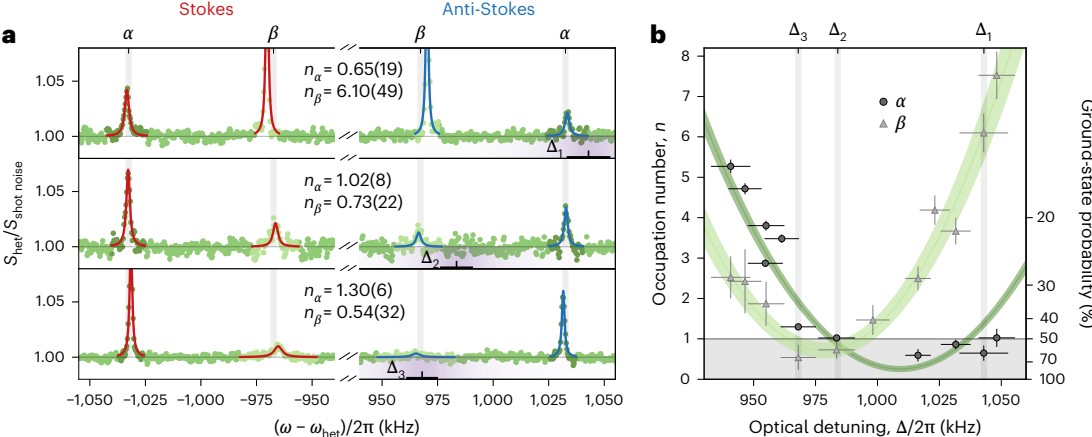

**Fig. 3 | Ground-state cooling of $\alpha$ and $\beta$ librations. a**, PSDs of detector 1 signal (green points) show cooling to the ground state of $\alpha$ (top) and $\beta$ (bottom), and close to the simultaneous ground state of $\alpha$ and $\beta$ (middle), for three different detunings of the cavity's $y$ mode ($\Delta_1, \Delta_2, \Delta_3$) = $2\pi \times (1{,}043, 984, 968)$ kHz. The cavity transfer function is indicated by the purple gradient at the bottom of each plot, with the central frequency and its $1\sigma$ uncertainty shown as a line marker. We extract the mode occupations from the imbalance between the fitted Stokes

(red) and anti-Stokes scattering (blue). **b**, Extracted phonon occupations of $\beta$ libration (grey triangles) and $\alpha$ libration (black circles) as a function of tweezer–cavity detuning $\Delta$. Theoretical predictions for occupations of the $\alpha$ and $\beta$ modes are shown as light and dark green regions, respectively. The green-shaded regions and the error bars denote $1\sigma$ uncertainties. The grey-shaded region marks a ground-state population with a probability greater than 50%.

spring and damper[52]. At the same time, the detuning strongly affects the cooling and final occupation number (Fig. 2e).

Fitting the curves shown in Fig. 2c,d allows us to extract the optomechanical coupling $g_\alpha$ (Methods). Since the geometry of the nanorotor determines the coupling strength, we can calculate the particle-specific moment of inertia about the $z$ axis as $I_b = 3.3 \pm 0.4 \times 10^{-32}$ kg m$^2$. In the coldest state ($n_\alpha = 0.21 \pm 0.03$), we determine the standard deviation of the librational amplitude as $\sigma_\alpha = 17.4 \pm 0.9$ µrad, corresponding to an effective temperature of $T_\alpha = 28 \pm 2$ µK (Methods).

## Ground-state cooling of two librational modes

To tightly align the particle with the polarization axis, we extend coherent scattering cooling now to both modes, $\alpha$ and $\beta$. A dumbbell formed from two silica spheres with $d = 156$ nm is trapped and oscillates at $\Omega_\alpha/2\pi = 1{,}035$ kHz and $\Omega_\beta/2\pi = 978$ kHz. As the two frequencies differ only by about twice the cavity decay rate $\kappa$, the two modes can be cooled simultaneously if the tweezer–cavity detuning is properly chosen and the phase-noise reduction is activated at both frequencies (Extended Data Fig. 2c).

We measure the occupation of both librational modes as a function of detuning $\Delta$ via sideband thermometry (Extended Data Fig. 3a,b). Because the librational modes $\alpha$ and $\beta$ couple to orthogonal cavity modes, we can treat their dynamics separately. By setting the detuning close to the librational frequencies $\Omega_\alpha$ or $\Omega_\beta$, we can cool the $\alpha$ or $\beta$ motion individually into their quantum ground states, namely, $n_\alpha = 0.65 \pm 0.19$ and $n_\beta = 0.54 \pm 0.32$, respectively (Fig. 3a). At $\Delta_2/2\pi = 984$ kHz, we achieve the lowest combined phonon number with $n_\alpha = 1.02 \pm 0.08$ and $n_\beta = 0.73 \pm 0.22$. This corresponds to effective librational temperatures of $T_\alpha = 73 \pm 4$ µK and $T_\beta = 57 \pm 28$ µK. Again, we fit the occupations as a function of detuning (Fig. 3b). The evaluation of optomechanical coupling from the oscillator linewidth (Methods and Extended Data Fig. 3c) reveals that the nanorotor's alignment along the $x$ axis is defined with an uncertainty of $\sigma_\alpha = 18 \pm 1$ µrad and $\sigma_\beta = 17 \pm 3$ µrad, which is close to the quantum zero-point fluctuations of about 13 µrad.

Such cooling near the quantum limit is an important prerequisite for future experiments on rotational interference and quantum sensing[21]. The aligned state corresponds to a coherent superposition of angular momentum states with a mean of $j \simeq \sqrt{k_B T I}/\hbar \approx 6 \times 10^4$. If we were to release the rotor non-adiabatically from its orientational ground state, it would evolve into a superposition of rotational quantum states with classically mutually exclusive angular momenta[27]. This

is expected to lead to rotational dispersion and quantum revivals due to the constructive interference of the rotational wave packets after a time $T_{rev} = 2\pi I/\hbar$. For the nano-dumbbell in our experiment, the revival time is 50 min. Therefore, observing revivals at a realistically observable timescale requires smaller particles or a scheme to resolve fractional revivals[28].

## Robust and repeatable trap loading and cooling

Advanced experiments in levitated optomechanics require sources that can load and cool nanoparticles with a high repetition rate and reliability, but are simultaneously able to handle different particle types and geometries. We demonstrate here the repeatable loading and ground-state cooling of half a dozen different nanorotors, formed from silica nanospheres with $d = 119$ nm. We limit this study to the ground-state cooling of $\alpha$ libration to save measurement time. The particles are coated on a glass slide covered by a 50-nm-thick silicon film and placed above the cavity (Fig. 4a). A green laser pulse with a duration of about 6 ns and an energy of 100 µJ focused down to a waist diameter of about 100 µm hits the backside of this sample and ejects the particles into dry nitrogen at a base pressure of 6 mbar (Extended Data Fig. 4 shows the experimental details). This can release single spheres, dumbbells, trimers or bigger clusters. The process is similar to laser-induced acoustic desorption (LIAD)[46], but because the absorption layer is thin enough to be fully evaporated, we refer to the method as laser-induced desorption (LID). As corroborated by scanning electron microscopy, the particles already aggregate in solution, but we also observe indications of occasional dimer growth in the trap through sequential capture of two spheres.

To characterize the geometry of the trapped rotor, we track its motion along the $x$ and $y$ axes by monitoring the transmitted tweezer light in a split detection scheme (Methods). Although the oscillator damping in the residual gas is the same along all axes for isotropic nanoparticles, the ratio of the damping rates $\gamma_y/\gamma_x$ was simulated to be 1.258 for dumbbells and 1.378 for linear trimers[14] (Fig. 4b).

After shape assessment at prevacuum, we activate the cavity and evacuate the system to high vacuum. As the particle shape approaches cylindrical symmetry, the system becomes increasingly susceptible to heating and mechanical instabilities due to resonances between $\gamma$ libration and the translational $z$ mode[40]. To stabilize $\gamma$, we introduce a slight ellipticity to the tweezer polarization during pump down. Once high vacuum is reached, we return to linear $x$ polarization; at this point,

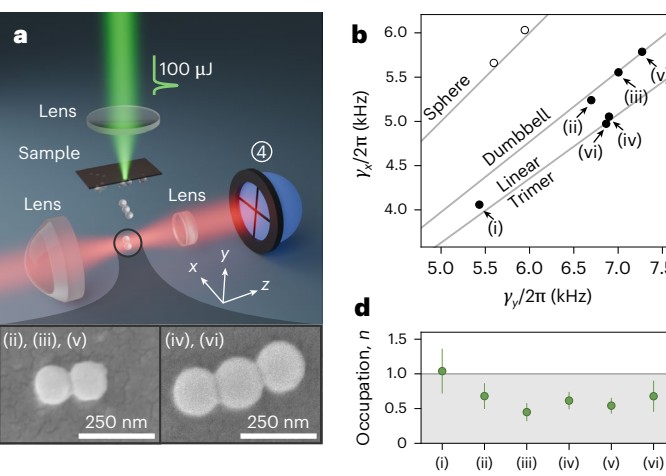

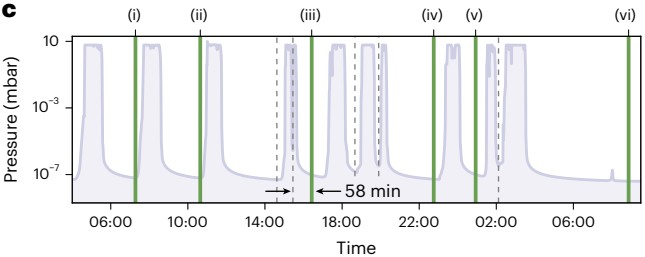

**Fig. 4 | Repeatable ground-state cooling. a,** Experimental sketch of loading nanorotors in vacuum with laser-induced desorption. A pulsed nanosecond laser is focused onto a sample coated with $SiO_2$ nanoparticles; scanning electron microscopy of such samples prepared with 119-nm particles reveals dumbbells and trimers (inset) alongside single spheres and agglomerated clusters. The ejected nanoparticles are trapped in the optical tweezer. With a split detection scheme in forward scattering (4, blue), the particle geometry is analysed by extracting the mechanical damping rates. **b,** Measured damping rates $\gamma_x$ and $\gamma_y$ for cluster (i), three dumbbells (ii), (iii) and (v), and two linear trimers (iv) and (vi), as well as simulated values for different geometries. **c,** Pressure trace during loading and ground-state cooling (highlighted by green solid lines). Particles that were ejected are highlighted as grey dotted lines. **d,** Measured occupation numbers of $\alpha$ libration. The error bars denote $1\sigma$ uncertainties. The grey-shaded region marks a ground-state population with a probability greater than 50%.

we maintain control even over axially symmetric dumbbells, although their $\gamma$ motion then remains largely free.

We have repeated the procedure of trapping, shape assessment, evacuation to high vacuum and cavity cooling for a series of nanoparticles over a period of ~28 h. Six nanorotors, marked (i)–(vi) in Fig. 4b–d, were successfully cooled (near) to their librational quantum ground state. They comprise dumbbells, trimers and clusters. To illustrate the scale of this experiment, we plot the trap pressure as a function of time (Fig. 4c). Every observation of librational ground-state cooling of a fresh particle is marked by a green line, whereas the grey dashed lines mark events of intentional or accidental particle loss (Extended Data Fig. 5 shows details about particle loss). The fastest cycle from the ejection of one particle to the ground-state cooling of a new one took 58 min, primarily limited by the duration of evacuation. The final occupation numbers for all six successful events are shown in Fig. 4d.

## Conclusion

Using our bimodal high-finesse cavity, we have demonstrated repeated ground-state cooling of the $\alpha$ and $\beta$ libration modes of differently shaped nanorotors via coherent scattering. We find occupation numbers down to $n_\alpha \simeq 0.21$ for a nanocluster composed of spheres with $d = 119$ nm and $n_\alpha \simeq 1.02$ and $n_\beta \simeq 0.73$ when optimizing the cavity detuning

for the simultaneous cooling of both librational degrees of freedom of a dumbbell with $d = 156$ nm. This process aligns the nano-dumbbell with an angular uncertainty close to its 2D quantum zero-point fluctuations.

Combined with our capability for fast loading and cooling of dumbbells, trimers and larger clusters, this is a stepping stone towards previously inaccessible tests of quantum mechanics and quantum-enhanced rotational torque sensing. However, there is a general trade-off between cooling efficiency and quantum readiness: larger particles are easier to manipulate and cool due to their higher polarizability and cavity coupling, whereas lighter particles exhibit faster wave-function expansion, facilitating both linear and rotational interferometry.

The nano-dumbbell used in our 2D cooling experiment, with a mass around $4 \times 10^9$ atomic mass units (u), would exhibit a rotational quantum-state revival time of $T_{rev} = 2\pi I/\hbar \simeq 1$ h, which is prohibitively long even when considering fractional revivals[56]. Rotational matter–wave interferometry will, therefore, require particles with smaller moments of inertia $I_{rot}$[27]. For a dumbbell made of two 20-nm silica spheres, with a total mass of $1 \times 10^7$ u, the revival time is 150 ms, corresponding to roughly 40 cm of free fall. This timescale is compatible with a laboratory-scale experiment, provided that particles with comparable moments of inertia can be prepared, or the same particle can be reused. Such an experiment is an important goal, as it could boost the macroscopicity value[57] by orders of magnitude beyond the current state of the art[58].

The mass scale of $1 \times 10^7$ u is also intriguing because it encompasses relevant nanobiological materials. The tobacco mosaic virus stands out as a natural nanorotor with a length of 300 nm, a diameter of 18 nm and a mass of $4 \times 10^7$ u. Such thermolabile materials will require soft loading and cooling methods, most probably in the dark[59]. Cooled to the ground state, a trapped tobacco mosaic virus would feature a resonant torque sensitivity of about $3 \times 10^{-29}$ N m Hz$^{-1/2}$ (ref. 60).

## Online content

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

## Methods

### Optical setup

The optical setup is shown in Extended Data Fig. 1. Light emitted by an infrared fibre laser (NKT Photonics Koheras Adjustik E15) passes through the fibre electro-optic modulator EOM 2. We split off a small fraction of the light to lock the cavity (Extended Data Fig. 1a). The rest is amplified to a power of 6 W (NKT Photonics Boostik HP) and then divided into three parts: one for phase-noise detection, one serving as the LO in heterodyne detection (Extended Data Fig. 1c) and up to 3 W for the optical tweezer.

The tweezer mode is cleaned by a polarization-maintaining fibre, and its polarization is set by wave plates to be linear along the cavity axis. This orientation minimizes Rayleigh scattering into the cavity when the nanorotors are perfectly aligned. The laser light fills the aspherical tweezer lens, which has a diameter of 25.4 mm, a numerical aperture of numerical aperture of 0.81 and an effective focal length of 13.2 mm (Thorlabs, custom design). For a cluster assembled using 119-nm nanospheres (Fig. 1b), we determine a trap power of $P = 2.7$ W and trapping waists of $w_x = 1.17$ μm and $w_y = 0.98$ μm.

We detect the trapped nanoparticle by collecting its backscattered light (Extended Data Fig. 1c). Its two polarization components are split by the PBS and detected separately. The vertical component provides the most information about the particle's rotation, particularly about the rotation around the tweezer propagation axis $z$. This signal is only weakly sensitive to Rayleigh scattering of the aligned rotor and scattering at surfaces along the beam path. Therefore, this component is used to monitor cooling to the librational ground state. The horizontal contribution is isolated using a fibre circulator, which provides intrinsic alignment of the backscattering signal and is, therefore, used during trap alignment. To reduce the Rayleigh scattering peak, we filter the electrical signal using a crystal oscillator.

The trapped nanoparticle is centred at an antinode of the cooling cavity mode. The resonator is formed using mirrors with intensity reflectivity of $R \geq 0.99999$ (FiveNine Optics) and radius of curvature of 5 cm, yielding a finesse of $\mathcal{F} \approx 300,000$ at a free spectral range of 9.72 GHz, corresponding to a linewidth of $\kappa/2\pi = 32.4$ kHz and a central waist of $w_{cav} = 94$ μm. By careful design of the cavity mirror mount (Extended Data Fig. 4), we achieve an alignment of the cavity modes both along and orthogonal to the direction of tweezer propagation. The birefringence splitting between the two modes can be tuned in the range of $2\pi \times (0–30$ kHz) by applying pressure onto the mirrors via screws.

We lock the laser to the cavity using the Pound–Drever–Hall scheme (Extended Data Fig. 1a). EOM 1 (iXblue, PHT MPZ-LN-10-00-P-P-FA-FA) generates the locking sidebands and is used together with acousto-optic modulator AOM 1 (G&H, T-M200-0.1C2J-3F2P) to shift the locking frequency by one free spectral range of the cavity. This minimizes interference between the locking and the cooling light in detection.

To detect the particle motion in all directions, we use a heterodyne scheme, which mixes the scattered light with an LO. This enhances the signal interferometrically and shifts the signal to a spectral range of lower noise. The LO is blueshifted by 4.99814 MHz with respect to the tweezer beam using two polarization-maintaining fibre modulators (G&H, T-M200-0.1C2J-3F2P): AOM 2 at 197.5 MHz and AOM 3 at −202.49 MHz (Extended Data Fig. 1c).

The scattered light transmitted by the cavity mirror is divided into its horizontal and vertical polarization components. They are individually combined with the LO beam using a 50:50 fibre beamsplitter (Thorlabs PN1550R5A2). Each polarization output is then detected by a balanced photodiode (Thorlabs PDB425C-AC). In both backplane detections, we use variable-ratio fibre beamsplitters (KS Photonics) to balance the outputs, which are also detected by balanced photodiodes (Thorlabs PDB440C-AC) (Extended Data Fig. 1c).

After the optical trap, we collimate the tweezer light using a low-numerical-aperture aspheric lens (Thorlabs C560TME-C)

and isolate the particle signal using a split detection scheme (Extended Data Fig. 1d). We use a D-shaped mirror to split the optical beam into two halves that are detected by balanced photodiodes (Thorlabs PDB440C-AC). This detection is built for both $x$ and $y$ axes.

### Phase-noise reduction

In the presence of the cavity, laser phase noise can heat the mechanical motion[55]. The cavity delays the release of scattered light, effectively creating an unbalanced interferometer in heterodyne detection between scattered light and the LO. The laser phase noise appears in cavity transmission as an increased noise background around the cavity mode resonance. In Fig. 1b, this is shown at a frequency of around ~800 kHz and fitted with a Lorentzian to extract the exact frequency and to determine the birefringence splitting. We also use the fitted frequency to determine the actual tweezer–cavity detuning and its error during the detuning scan (Fig. 2e).

Strong cooling of the librational modes without the active suppression of phase noise leads to noise squashing[61] (Extended Data Fig. 2c, top), which distorts the motional sideband and generates a dip in the phase-noise background. This prevents accurate sideband thermometry. We, therefore, implement a phase-noise reduction scheme, using an unbalanced Mach–Zehnder interferometer[43] (Extended Data Fig. 1b). The short arm contains a polarization-maintaining fibre attenuator to equalize the optical power in both arms. The long arm consists of a 100-m single-mode fibre (SMF-28), enclosed in a chamber at prevacuum. This arm also includes a fibre stretcher to stabilize slow path-length fluctuations (>10 ms), and it combines a manual fibre polarization controller and a fibre PBS to correct for polarization changes. Light from both arms is recombined using a 50:50 fibre coupler and directed to a balanced detector (Thorlabs PDB450C-AC). After filtering, the interferometer output is fed back into EOM 2, which controls the phase of the tweezer light.

With active feedback, the noise level is reduced by more than 30 dB both at a single frequency (Extended Data Fig. 2b) and two frequencies (Extended Data Fig. 2c, bottom). The reduction is also visible in cavity transmission, restoring the expected shape of the motional sidebands (Extended Data Fig. 2, middle).

### Mode identification

To assign the peaks shown in Fig. 1b to translational and librational modes, we first use the fact that the translational frequencies for nanoparticles much smaller than the optical wavelength hardly depend on the particle shape. We, therefore, use individual spherical nanoparticles to identify the frequencies associated with the $z$, $x$ and $y$ modes, where the $x$ and $y$ frequencies change depending on the tweezer polarization, whereas the $z$ frequency stays invariant. When switching to anisotropic nanoparticles, three additional frequency peaks appear. Due to the prolate geometry of our nanorotors (mostly dimers and linear trimers), we have one peak at smaller frequencies ($\gamma$) and two peaks at larger frequencies. As described in the 'Experimental setup' section, we use the polarization-sensitive detection of the cavity transmission to discriminate between $\alpha$ and $\beta$.

### Theoretical description

The nanorotor is an asymmetric rigid body ($I_c < I_b < I_a$), whose orientation in the laboratory frame ($\mathbf{e}_x$, $\mathbf{e}_y$, $\mathbf{e}_z$) is specified by the three Euler angles ($\alpha$, $\beta$, $\gamma$), using the $z–y'–z''$ convention (Fig. 1a, inset). Its optical response is characterized by the susceptibilities $\chi_a < \chi_b < \chi_c$ (ref. 42), which can be combined into the susceptibility tensor $\chi = \chi_a \mathbf{n}_1 \otimes \mathbf{n}_1 + \chi_b \mathbf{n}_2 \otimes \mathbf{n}_2 + \chi_c \mathbf{n}_3 \otimes \mathbf{n}_3$, where $\mathbf{n}_1$, $\mathbf{n}_2$, $\mathbf{n}_3$ are basis vectors. The particle is illuminated by the linearly polarized tweezer field $\mathbf{E}_{tw}(\mathbf{r}) = E_{tw}(\mathbf{r})\mathbf{e}_\phi$ of wavelength $2\pi/k$ with the tweezer mode amplitude $E_{tw}(\mathbf{r}) \propto e^{ikz}$ propagating in the $\mathbf{e}_z$ direction and the polarization direction $\mathbf{e}_\phi = \mathbf{e}_x \cos\phi + \mathbf{e}_y \sin\phi$. Coherent scattering of tweezer photons couples the deeply trapped particle rotations to two orthogonally polarized modes of the cavity

field $\mathbf{E}_c(\mathbf{r}) = E_c(\mathbf{r})(\mathbf{e}_y a_y + \mathbf{e}_z a_z)$, with $E_c(\mathbf{r}) \propto \cos(kx)$ denoting the cavity mode amplitude and $a_{y,z}$ the corresponding complex mode variables. The resulting interaction potential can be derived from the Lorentz torque acting on the particle as[35,36,42]

$$U = -\frac{\varepsilon_0 V}{4} \mathbf{E}_{\text{tw}} \cdot \chi \mathbf{E}_{\text{tw}}^*$$
$$-\frac{\varepsilon_0 V}{4} (\mathbf{E}_c \cdot \chi \mathbf{E}_{\text{tw}}^* + \text{h.c.}). \tag{4}$$

Here $V$ denotes the particle volume and $\mathbf{R}$ is the particle centre-of-mass position. Since the particle remains stably trapped at $\mathbf{R} \simeq 0$, the first term describes the librational trapping near $(\alpha, \beta) \simeq (\phi, \pi/2)$. The second term describes the coupling of librations in $\alpha$ and $\beta$ to two orthogonally polarized cavity modes as well as trapping of $\gamma$. In our experiment, $\gamma \simeq 0$ or $\gamma \simeq \pi/2$ because the cavity modes are polarized along $\mathbf{e}_y$ and $\mathbf{e}_z$. In the following, we assume $\gamma \simeq \pi/2$; the case of $\gamma \simeq 0$ can be obtained by exchanging indices $a \leftrightarrow b$. The librational frequencies for deviations of $\alpha$ and $\beta$ from their equilibrium orientation are given by

$$\Omega_\alpha = \sqrt{\frac{\varepsilon_0 V}{2 I_b}(\chi_c - \chi_a)}|E_{\text{tw}}(0)|,$$
$$\Omega_\beta = \sqrt{\frac{\varepsilon_0 V}{2 I_a}(\chi_c - \chi_b)}|E_{\text{tw}}(0)|. \tag{5}$$

The second term in equation (4) decomposes into an orientation-independent term that drives the in-plane cavity mode $a_y$ and an orientation-dependent term that describes coupling between the cavity modes and particle librations. Specifically, the former term can be written in the form $V_{\text{dr}} = \hbar(\eta a_y + \text{h.c.})$ with the pump rate

$$\eta = -\frac{\varepsilon_0 \chi_a V}{4\hbar} E_c(0) E_{\text{tw}}^*(0) \sin\phi. \tag{6}$$

Likewise, the coupling between librations and the cavity modes follows from the orientation of the susceptibility tensor. For $\phi = 0$, the coupling becomes approximately linear in both librational degrees of freedom:

$$U_{\text{int}} \approx k_\alpha \alpha a_y + k_\beta \beta a_z + \text{h.c.}, \tag{7}$$

where the complex-valued constants for both librational modes are given by

$$k_\alpha = \frac{\varepsilon_0 V}{4}(\chi_c - \chi_a) E_c(0) E_{\text{tw}}^*(0),$$
$$k_\beta = \frac{\varepsilon_0 V}{4}(\chi_c - \chi_b) E_c(0) E_{\text{tw}}^*(0). \tag{8}$$

For both values of $\gamma$, equation (7) shows that $\alpha$ couples to the in-plane cavity mode $a_y$, whereas $\beta$ couples to the out-of-plane cavity mode $a_z$. Cavity-transmission spectra (Fig. 1b) consistently show $\Omega_\alpha > \Omega_\beta$ across all nanorotors trapped in our setup, which is compatible with $\gamma \simeq \pi/2$ and motivates this choice in our modelling.

We define the librational mode variables $b_\alpha = \alpha_{\text{zpf}}(\alpha + ip_\alpha/I_a\Omega_\alpha)$ and $b_\beta = \beta_{\text{zpf}}(\beta - \pi/2 + p_\beta/I_\beta\Omega_\beta)$, with zero-point fluctuation amplitudes $\alpha_{\text{zpf}} = \sqrt{\hbar/2 I_b \Omega_\alpha}$ and $\beta_{\text{zpf}} = \sqrt{\hbar/2 I_a \Omega_\beta}$, to obtain the quantized interaction Hamiltonian of equation (1), where we introduced the coupling constants

$$g_\alpha = \alpha_{\text{zpf}} k_\alpha,$$
$$g_\beta = \beta_{\text{zpf}} k_\beta. \tag{9}$$

In summary, this leads to the total libration cavity Hamiltonian in equation (2). A standard calculation then yields the optomechanical damping rates and the resulting steady-state occupation in equation (3)[42].

## Optomechanical coupling

The optomechanical coupling determines the interaction between the particle and cavity mode and, therefore, the cooling performance. By solving the equations of motion, with cooling providing additional damping, we obtain an effective motional linewidth of

$$\gamma_\mu^{\text{eff}}(\omega) = \gamma_\mu + \frac{4|g_\mu|^2 \Omega_\mu \Delta_c \kappa}{\left[\left(\frac{\kappa}{2}\right)^2 + (\omega + \Delta_c)^2\right]\left[\left(\frac{\kappa}{2}\right)^2 + (\omega - \Delta_c)^2\right]}, \tag{10}$$

which depends on the coupling strength. In the regime of strong cooling, when the cavity resonance is close to the mechanical frequency, energy loss through the cavity determines the damping and the cavity-induced linewidth dominates over the thermal linewidth $\gamma_\mu$. We use this expression to fit the linewidths extracted from cavity-detuning scans for 1D (Fig. 2d) and 2D (Extended Data Fig. 3c) cooling with a constant coupling. We verify the extracted coupling by additionally fitting the observed optical spring effect (Fig. 2c):

$$\Omega_\mu^{\text{eff}}(\omega) = \sqrt{\Omega_\mu^2 - \frac{4|g_\mu|^2 \Omega_\mu \Delta_c \left[\left(\frac{\kappa}{2}\right)^2 - \omega^2 + \Delta_c^2\right]}{\left[\left(\frac{\kappa}{2}\right)^2 + (\omega + \Delta_c)^2\right]\left[\left(\frac{\kappa}{2}\right)^2 + (\omega - \Delta_c)^2\right]}}. \tag{11}$$

Since the optomechanical coupling is determined by the rotor geometry, we can determine the moment of inertia for each mode. Combining equations (5) and (8) with the zero-point fluctuation, we calculate as follows:

$$I_b = \frac{|g_\alpha|^2}{\Omega_\alpha^3} \frac{|E_{\text{tw}}(0)|^2}{|E_c(0)|^2 8\hbar}, \quad I_a = \frac{|g_\beta|^2}{\Omega_\beta^3} \frac{|E_{\text{tw}}(0)|^2}{|E_c(0)|^2 8\hbar}. \tag{12}$$

## Noise contributions

For quantum-limited measurements, the signal must be isolated from noise. The noise contributions in backscattering detection are shown in Extended Data Fig. 2a. The raw spectrum contains dark noise (photodetector and oscilloscope), shot noise and phase noise of the LO. The latter originates from the frequency generators that drive LO AOMs 2 and 3 (Extended Data Fig. 1c). In postprocessing, we, therefore, subtract the background levels as extracted from the Lorentzian fits. Additionally, the detector sensitivity shows a weak frequency dependence, which differs for the Stokes and anti-Stokes peaks. The sensitivity is calibrated by acquiring the spectra of dark noise and LO's shot noise. Since shot noise is white, any residual frequency dependence must be due to the detector response. We, therefore, divide the background-corrected signals by the difference between shot noise and dark noise.

## Occupation number

The areas of the Stokes ($A_S$) and anti-Stokes ($A_{aS}$) peaks scale with the mean occupation number $n$ of the harmonic oscillator as $A_S = C(n + 1)$ and $A_{aS} = Cn$, respectively, where $C$ is a proportionality constant. The occupation number can, therefore, be extracted from the ratio of the Stokes and anti-Stokes peak areas[62]. In practice, the precision of area measurements is limited by the available integration time. When recording a detuning scan within a fixed total acquisition time, increasing the number of detuning points necessarily reduces the integration time per point, which would, in turn, degrade the precision of the occupation number estimates. Since the difference in the sideband areas satisfies $A_S - A_{aS} = C$, independent of the occupation number $n$, we determine $C$ by averaging the differences $A_S - A_{aS}$ over all spectra in a given scan. Figure 2b and Extended Data Fig. 3a,b show the resulting normalized peak areas $A_S/C$ and $A_{aS}/C$, whose difference is supposed to be unity by construction. The occupation number at each detuning is then obtained from $n = (A_S + A_{aS} - C)/2C$. With this procedure,

the statistical uncertainty of each extracted $n$ is comparable with the uncertainty obtained by spending the entire integration time on a single detuning point. In other words, pooling the area differences across the full scan allows us to estimate $n$ with high precision and still resolve its detuning dependence.

Knowing $n$, we estimate the mode temperature $T$ by assuming the Bose–Einstein distribution for a quantum harmonic oscillator in thermal equilibrium:

$$T = \frac{\hbar\Omega_\mu}{k_B}\left(\ln\left[1 + \frac{1}{n}\right]\right)^{-1}.$$  (13)

From the same thermal distribution, we also extract the ground-state population probability as

$$p_0 = 1 - \exp\left(-\frac{\hbar\Omega_\mu}{k_B T}\right) = \frac{1}{1+n}.$$  (14)

### Heating rates

In the absence of external heating, cooling is governed by the cavity-enhanced imbalance between anti-Stokes and Stokes scattering. For both processes, we define the weak-coupling damping and heating rate as

$$A_\mu^\pm = \frac{|g_\mu|^2\kappa}{(\kappa/2)^2 + (\Delta \pm \Omega_\mu)^2},$$  (15)

which yields, together with equation (3), a minimum occupation number of $n_{min} = \kappa^2/16\Omega_\mu^2$. It depends only on the cavity linewidth and mechanical frequency. For librational frequencies of $\sim 2\pi \times 1$ MHz, this implies a theoretical lower bound of $n_\alpha \approx n_\beta \approx 6.2 \times 10^{-5}$, far below our measured values. The system must, therefore, be limited by other sources, such as recoil heating, gas collisions or phase noise.

The recoil limit depends on both cavity and tweezer parameters. For our linearly polarized tweezer, we estimate $\Gamma^{recoil} = 3.2$ kHz (ref. 42), which limits cooling to $n_{recoil} = 0.064$. Phase noise and collisional contributions, however, vary with the particle geometry, as this determines the librational frequency and collisional cross-section. The phase-noise occupation can be obtained using equation (3).

We analyse heating and decoherence for the ground-state-cooled nanocluster (Fig. 2); the frequency dependence of the occupation reveals that the phase-noise contribution of $n_\phi(\Omega_\alpha) = 0^{+0.01}$ is negligible. Additionally, the fit displays a total heating rate of $\Gamma_\alpha = 6.8 \pm 0.7$ kHz, originating from both recoil and thermal noise. Since the former is pressure independent, the thermal part follows by subtraction from the total heating rate $\Gamma_\alpha^{thermal} = 3.6 \pm 0.8$ kHz. For this cluster particle, recoil and thermal heating contribute approximately equally.

The same noise analysis can be performed for the trapped nano-dumbbell, where we treat both librational modes separately (Fig. 3). For $\beta$ libration, the fit finds the phase noise to dominate with an occupation of $n_\phi(\Omega_\beta) = 0.38 \pm 0.17$, whereas the $\alpha$ mode is again only barely affected by it, with $n_\phi(\Omega_\alpha) = 0^{+0.07}$. From the fitted total heating rates in both dimensions, namely, $\Gamma_\beta = 20 \pm 4$ kHz and $\Gamma_\alpha = 18 \pm 2$ kHz, we estimate the thermal heating rates as $\Gamma_\beta^{thermal} = 16 \pm 4$ kHz and $\Gamma_\alpha^{thermal} = 14 \pm 2$ kHz, respectively. We conclude that collisional heating dominates the $\alpha$ mode, whereas $\beta$ libration is also limited by phase noise.

## Data availability

The data underlying the figures are available from the University of Vienna PHAIDRA repository at https://doi.org/10.25365/phaidra.770.

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

## Acknowledgements

We thank K. Hornberger, I. Coroli and N. Gupta for useful discussions during the planning and construction of the experiment and S. Puchegger and the Faculty Center for Nano Structure Research at the University of Vienna for support in SEM imaging. We thank the nanorotors 'Neal Particleton' and 'Quinn Tumman', who stayed in the trap for many days, for providing the first impressive data on librational cooling in this experiment and all other nanorotors for their repeated visits to our trap. This work relates to Department of Navy award N62909-23-1-2029 issued by the Office of Naval Research. S.T. acknowledges support from the Austrian Academy of Sciences (ÖAW) through an ESQ discovery project. B.A.S. acknowledges funding by the Carl Zeiss Foundation through the project QPhoton and by the DFG-510794108. U.D. acknowledges: this research was funded in whole or in part by the Austrian Science Fund (FWF) 10.55776/STA175.

## Author contributions

S.T. conceived the experiment with support from F.F., L.H., M.A. and U.D., based on theoretical predictions from H.R. and B.A.S. S.T., F.F. and L.H. built the experiment. S.T. and F.F. acquired and analysed the data. B.A.S., U.D. and M.A. supervised the project. S.T., F.F., B.A.S., U.D. and M.A. wrote the manuscript. All authors were involved in reviewing the draft.

## Funding

## Competing interests

The authors declare no competing interests.

## Additional information

**Extended data** is available for this paper at https://doi.org/10.1038/s41567-026-03219-1.

**Correspondence and requests for materials** should be addressed to Benjamin A. Stickler or Markus Arndt.

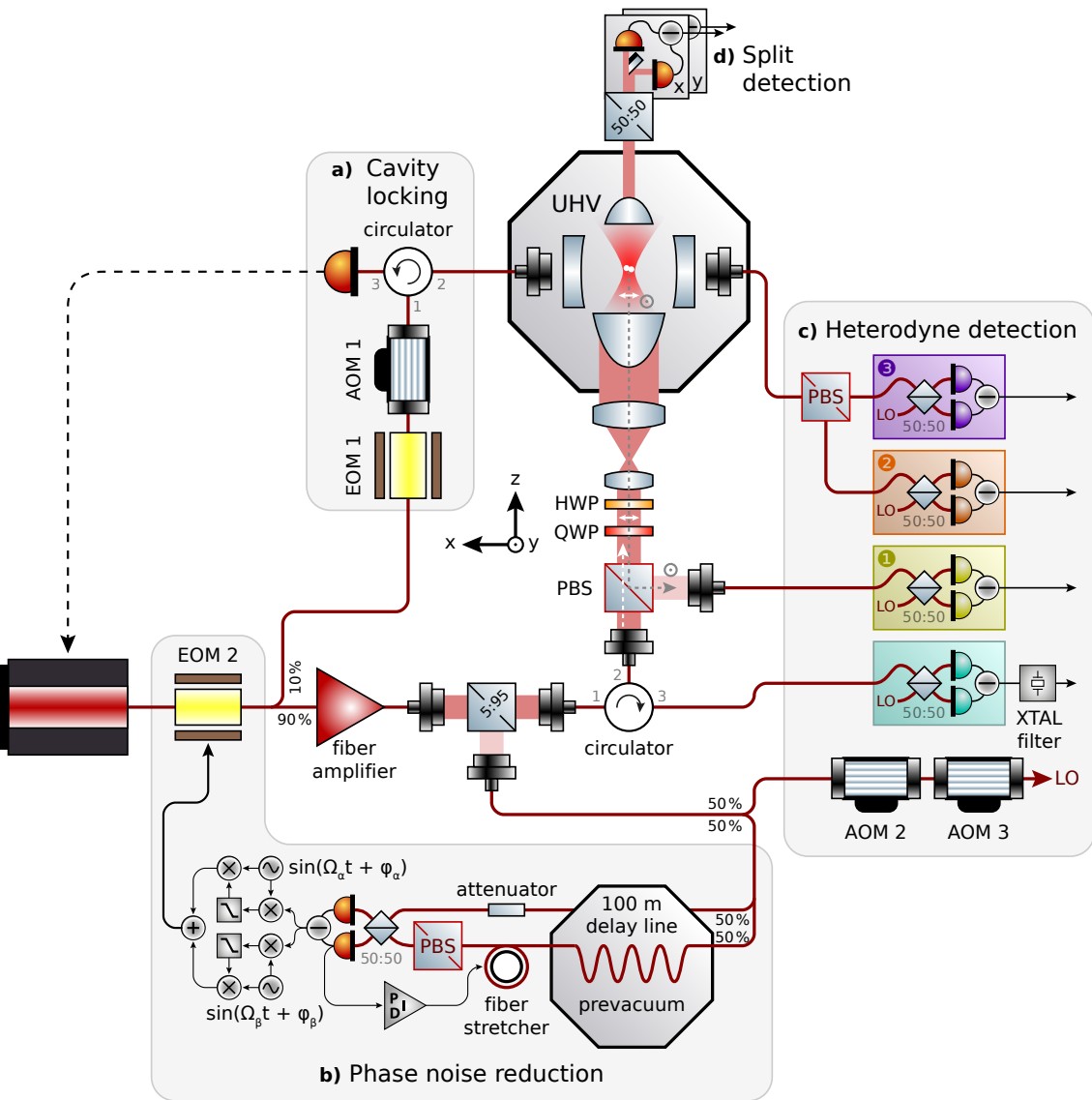

**Extended Data Fig. 1 | Extended optical setup for trapping, cooling and detection.** The fiber laser beam is modulated by the electro-optic modulator EOM 2 to reduce its phase-noise level before it is amplified. The trapping beam is polarization-controlled by a quarter- (QWP) and half-wave plate (HWP), expanded and re-focused into a sub-micron waist at the center of an antinode of the cooling cavity. **a**) Pound-Drever-Hall locking of the laser to the cooling cavity, shifted by about one free spectral range, that is by 9.72 GHz (shifted by EOM 1 and acousto-optical modulator AOM 1). The cavity frequency is additionally blue-detuned with respect to the tweezer light by about one mechanical eigenfrequency of the libration that should be cooled, that is, by 0.5–1.3 MHz. **b**) Unbalanced Mach-Zehnder interferometer with 100 m fiber loop to measure laser phase noise, in particular sensitive at the librational frequencies around 1 MHz off the infrared carrier frequency. A fiber stretcher is used to stabilize the length of both arms. The recombined arms are monitored on a balanced detector, the output of which is used to actively compensate the phase noise at both librational frequencies using EOM 2. **c**) Polarization-sensitive heterodyne detection. Part of the laser light gets up- and down-shifted by AOM 2 and AOM 3 to prepare the local oscillator (LO) for heterodyne detection about 5 MHz blue-detuned with respect to the tweezer light. Heterodyne detection ① of the y-polarized light in the back-plane is used to monitor all motional degrees of freedom of the nanorotor using the light that is backscattered by the particle, which is mixed with the LO on the photo-detector. An additional heterodyne detector is used to monitor the x-polarized light via a fiber circulator. A notch filter based on a crystal oscillator (XTAL) is used to reduce the signal level of Rayleigh scattered light. The cavity transmission is used to distinguish both librational modes $\alpha$ and $\beta$ and the cavity mode frequencies on balanced detectors ② and ③. **d**) The transmitted tweezer light is collimated and halved by a beam splitter. The two halves are fed to two split detection schemes; one sensitive to the motion along the x-axis, one along the y-axis.

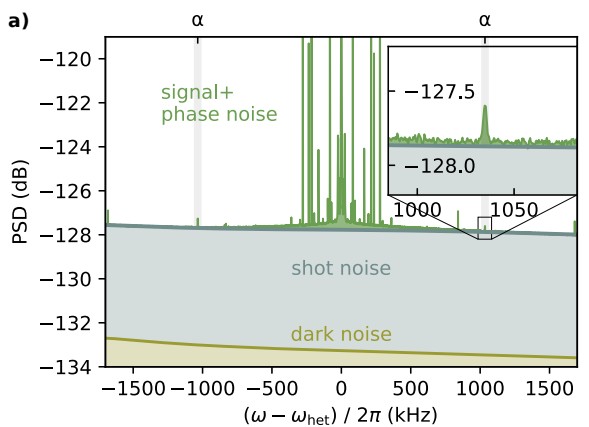

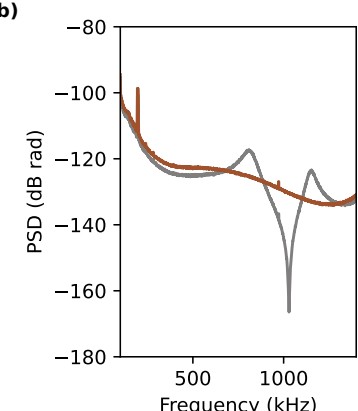

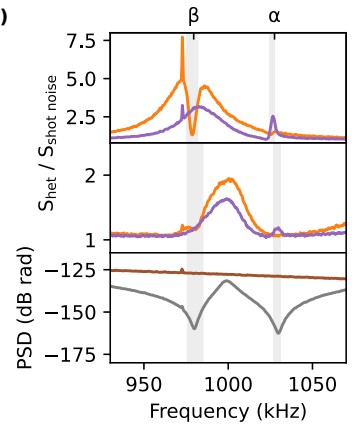

**Extended Data Fig. 2 | Characterization of the principal noise components. a)** Power spectral density of the particle motion, as measured by back-scattering heterodyne detection ①. The $\alpha$-libration appears at a frequency of $2\pi \times 1030$ kHz and albeit rather weak, it can be clearly discerned because most noise components appear at lower frequency. The strong peaks below 400 kHz are due to the linear motion along the three linear degrees of freedom, x, y and z, as well as some higher harmonics and beat notes. Blocking all laser light provides the detector and oscilloscope dark noise. Blocking the signal and thus detecting the local oscillator provides the shot noise level. **b)** Measured phase-noise curve with an unbalanced Mach-Zehnder interferometer. The brown line shows the measured phase noise for relevant particle frequencies. With active feedback on

EOM 2, the phase noise is reduced around the $\alpha$-libration by 35 dB. This is crucial for cooling the nanorotor into its 1D librational quantum ground state. **c)** Dual-mode phase-noise reduction for 2D ground-state cooling. In cavity transmission (output of detectors ② and ③), we observe the distortion of the motional peaks to a dip in the signal by noise squashing (top panel). When reducing the phase noise around the motional frequencies the phase-noise pile-up is suppressed and the motional peaks become visible (center panel). We achieve a phase-noise reduction of 30 dB around the $\alpha$- and $\beta$-mode with a frequency separation of $2\pi \times 50$ kHz simultaneously (bottom panel), which enables ground-state cooling of both modes separately, as well as cooling close to the 2D ground state.

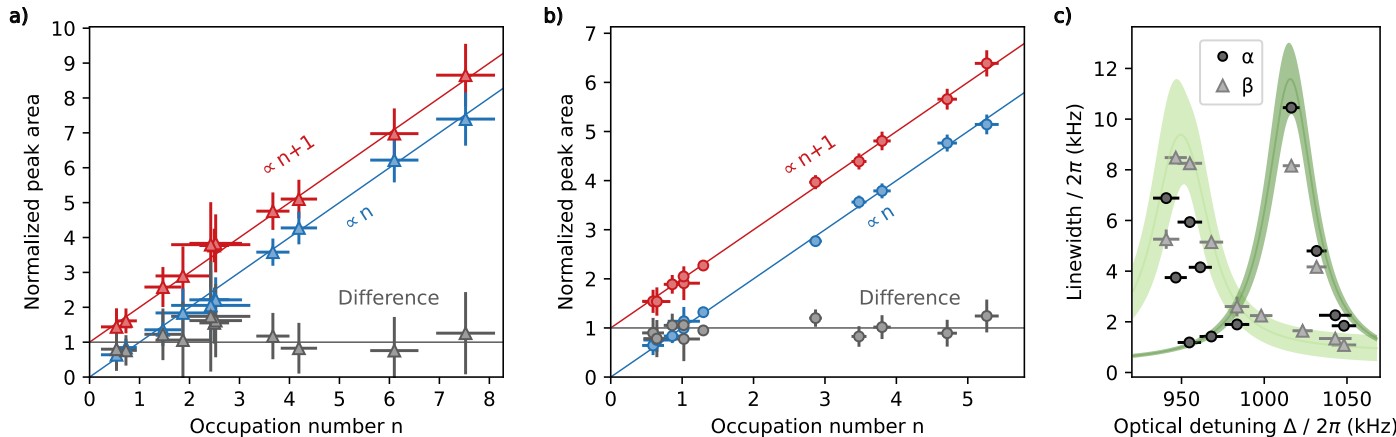

**Extended Data Fig. 3 | Evaluation of phonon occupation and optomechanical coupling strength for 2D cooling. a)–b)** The areas of the Stokes (red markers and linear fits) and anti-Stokes peaks (blue markers and linear fits) are normalized to their respective differences, for both (a) $\beta$ and (b) $\alpha$. Based on the peak areas, we determine the occupation number n (see Methods). **c)** Optical damping due to the cavity coupling: Sweeping the detuning $\Delta$ between the cavity and the tweezer

frequency allows us to scan across the mechanical resonance of both librational modes. We show the fitted linewidth (green) for the $\beta$-mode (gray triangles) and the $\alpha$-mode (black circles). The green shaded areas denote the $1\sigma$ uncertainty from fitting. Due to experimental instabilities of the trapping frequency, some linewidth measurements show excessive values, which were excluded from the fit. Error bars denote $1\sigma$ uncertainties.

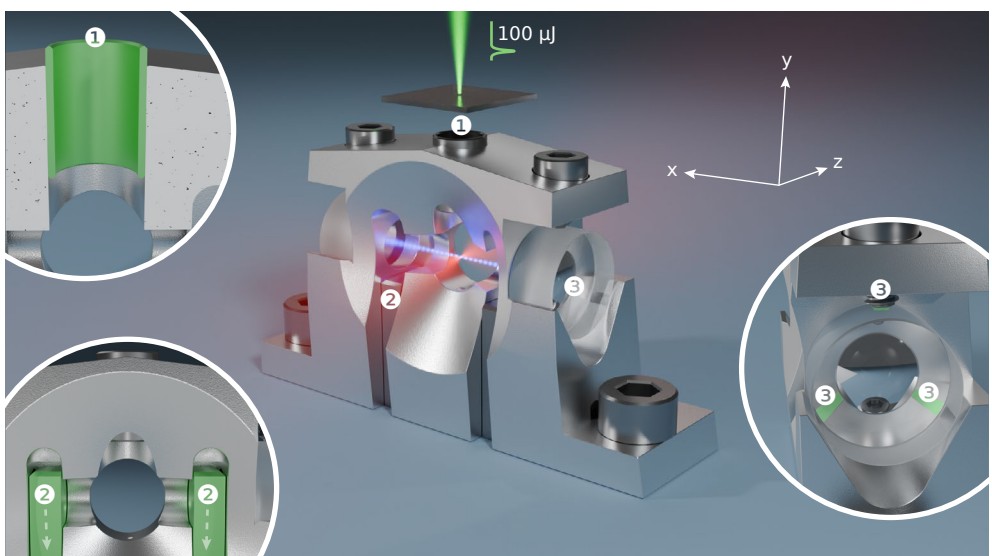

**Extended Data Fig. 4 | Experimental details of the trapping site.** A pulsed nanosecond laser ($E_{pulse} = 100\ \mu J$) is focused onto a cover slip coated with a 50 nm thick layer of silicon and $SiO_2$ nanoparticles, both on the bottom side. The particles diffuse in 6 mbar of dry nitrogen to the focus of the optical tweezer where they are trapped. Over thousands of shots, charged material can accumulate and electrostatically clog the upper hole of the mount, reducing the efficiency of trap loading. A simple remedy to this is a metallic tube ① in the mount that can be easily exchanged. In order to shield the cavity mirrors from contamination they are protected by a retractable shield ② during the loading process. The cavity mirrors are glued onto the Invar holder. A screw with a soft tip applies an adjustable force on the top of the mirror ③. Together with contact areas separated by 120°, a controlled birefringence is induced in the cavity mirror coating, which leads to a cavity mode splitting along the y- and z-axis.

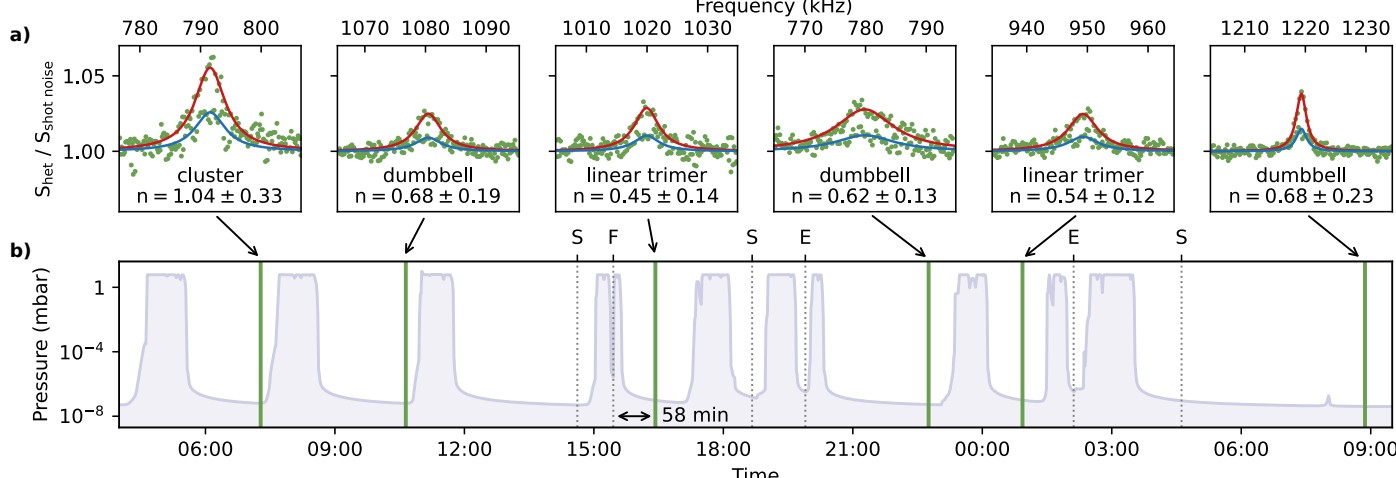

**Extended Data Fig. 5 | Ground-state cooling of different nanorotors trapped in the same setup on the same day. a)** PSDs of the successfully cooled nanorotors, where the red fit corresponds to the Stokes and the blue fit to the anti-Stokes peak. The particles were LID-loaded and characterized at 6 mbar, evacuated to high vacuum, and cooled by coherent scattering. We demonstrate ground-state cooling of the $\alpha$-motion for three dumbbells and two linear trimers as well as one cluster. **b)** The pressure trace during the measurement time acts as a timeline, where the successfully cooled particles are marked by vertical green lines. The time stamps of the lab clock show the repeatability of the process. The dashed line marked with 'F' indicates that the librational frequency was too low for ground-state cooling. The marker 'E' indicates an operational error. The marker 'S' describes when the $\gamma$-libration heated the other degrees of freedom too much and destabilized the particle orientation. As the last particle demonstrates, ground-state cooling is nevertheless possible after a longer waiting time.

