## [Peer Review file · Nature Physics]

Quantum ground-state cooling of two librational modes of a nanorotor

Corresponding Author: Professor Markus Arndt

Version 0:

Reviewer comments:

Reviewer #1

(Remarks to the Author)
See attachment.

Reviewer #2

(Remarks to the Author)

The paper “Quantum ground-state cooling of two librational modes of a nanorotor” reports the simultaneous quantum ground-state cooling of two librational modes of a levitated silicon nanorotor held in an optical dipole trap. This work extends the paradigm of levitated optomechanics—traditionally focused on the center-of-mass motion of spherical nanoparticles—to rotational degrees of freedom inherent to nonspherical objects. Such systems possess rich internal dynamics, including torsional and librational modes, that can couple to optical and mechanical fields in ways unavailable to spheres. Demonstrating quantum control over these internal degrees of freedom represents a major step toward exploring the quantum mechanics of rotation and orientation. It opens perspectives for fundamental studies of macroscopic quantum phenomena, precision torque sensing, and the realization of hybrid quantum systems exploiting quantized rotational motion.

The authors present a well-designed experimental setup in which an anisotropic silicon nanorotor is optically trapped and coupled to a high-finesse cavity field. They map the system dynamics onto a standard optomechanical model, treating the two librational modes as harmonic oscillators interacting with the cavity via radiation-pressure-like coupling. The paper first demonstrates resolved-sideband cooling of a single librational mode to near its quantum ground state, followed by simultaneous cooling of two orthogonal modes, supported by calibrated noise spectra and thermometry analysis. The experimental protocol includes an automated trap-loading and feedback-cooling sequence enabling stable operation over a 28-hour run, attesting to the robustness of the setup. I appreciate both the clarity of presentation and the thoroughness of the data analysis. The achieved quantum control of two rotational modes constitutes an important milestone for optomechanics with nanorotors. Whether the work merits publication in Nature Physics will, however, depend on the resolution of several specific issues discussed below.

Major points

1. Scope of the advance

Cooling of one librational mode has already been demonstrated by Dania et al. [Ref. 40]. The present work extends this to two dimensions. In how far does this represent a substantive conceptual advance rather than a technical extension? What specific difficulties prevented earlier experiments from reaching this regime, and what concrete obstacles were overcome here? This is a critical aspect—the answer must convincingly establish the significance of the advance.

2. Motivation

The motivation in the abstract and introduction appears rather weak. What exactly are the “plethora of phenomena and applications” the authors refer to? This should be stated more concretely. Moreover, the phrase “...are only starting to receive the attention they deserve” sounds somewhat defiant and should be reformulated in a more neutral scientific tone.

3. Mode identification (Section II)

In Section II, the authors present spectral densities and identify the observed peaks as corresponding to specific librational and rotational modes of the nanorotor. What is the exact procedure underlying this mode assignment? Is it purely based on comparison with ab initio theoretical modelling—using predicted mode frequencies and light-scattering selection rules—or are there independent experimental checks? How accurate are the theoretical frequency predictions, and how sensitive are they to uncertainties in particle geometry, trap alignment, or optical parameters? Clarifying how reliably peaks are assigned to particular modes is essential for evaluating the robustness of the analysis.

4. Experimental routine (Section IV)

Section IV describes an impressive 28-hour experimental run, but the experimental procedure is explained too briefly. A clearer account of the trapping, evacuation, and cooling sequence would be helpful.

Smaller issues

Introduction

- 5) Explain why rotational wave functions can be “divided up and recombined naturally,” in contrast to translational wave functions.
- 6) Give a clearer account of the accessible mass range in nanorotor experiments. What is meant by a “small molecule”? The challenge appears to be reaching the MDa–GDa range, yet the conclusions suggest that smaller masses are needed to observe revivals. What is the optimal mass range for quantum experiments, and what limits it? This point needs to be clarified also in the conclusions.
- 7) Clarify what is meant by “a state that is aligned at the quantum level” and “the nanorotor’s orientation is defined at the quantum limit.”

Section I

- 8) The optomechanical coupling constants g are not explained in the main text.
- 9) The two librations couple selectively to the polarizations. While this is emphasized as remarkable, the authors do not explain it.
- 10) Clarify the phrase “According to the theoretical coupling.”
- 11) The explanation of cavity cooling seems overly complicated. What is meant by “At resonance, the cavity enhances the spectral mode density”?
- 12) Before Eq. (3), what is Γ ? Should this be Γ_{μ} ? Perhaps remind the reader that $\mu = \alpha, \beta$, as must be inferred from Eq. (2).
- 13) The remarks on laser stabilization and background gas collisions could be moved to the appendix unless they are essential for understanding the data, which does not seem to be the case.

Section II

- 14) Specify which objects are cooled.
- 15) The notation A^{\pm} and $A_{S/aS}$ for scattering rates and PSD areas is somewhat confusing.

Section IV

- 16) In Fig. 4b, the grey lines represent theoretical expectations. Does their thickness correspond to uncertainties?

The paper addresses a timely and important problem and presents technically demanding experiments with strong potential impact. The results are impressive, but the above points—particularly the justification of the conceptual advance (1) and the clarification of experimental and analytical details (3, 4)—should be convincingly addressed before the work can be considered for publication in Nature Physics.

Version 1:

Reviewer comments:

Reviewer #1

(Remarks to the Author)
See attached.

Reviewer #2

(Remarks to the Author)

I have carefully read both referee reports, the authors’ response, and the revised manuscript. Overall, I am satisfied with the authors’ replies to the questions and comments raised in the first round, and with the corresponding changes made to the manuscript. In both reports, the most critical issue concerned the novelty of the present work in comparison to previous studies. While the extension from cooling a single librational mode to two modes still appears, at first sight, somewhat incremental, the authors convincingly argue that this step is essential for accessing and exploiting genuine nanorotor physics, which naturally lives in a two-dimensional phase space.

In the revised manuscript, this motivation is presented more clearly, and I largely follow and agree with the authors’ line of argumentation. The comparison with previous literature has been improved, and both the style and tone of the presentation are more balanced and precise. One point, however, still merits attention. I was surprised by the paragraph starting with “Here, we demonstrate ...” on page 1. This statement is noticeably weaker than the case made in the authors’ response to the referees and reflects only part of the advances discussed there. In particular, progress regarding particle loading and characterization, as well as improvements in trap stability, are not mentioned at all. If the article is intended to demonstrate key enabling steps towards nanorotor optomechanics, I would recommend that all of these advances be clearly reflected already in the introduction, so that the scope and significance of the work are immediately apparent to the reader. This remark is meant as an optional suggestion for further strengthening the presentation.

Overall, I am satisfied with the revised manuscript and recommend acceptance.

Reply to the Reviewers

We thank both referees for their time and supportive feedback, and for emphasizing the importance of more clearly positioning our work within the broader progress in levitated optomechanics. In our enthusiasm for rotational physics, the first submission had not yet sufficiently acknowledged earlier achievements in other degrees of freedom. We have now reworked and expanded the introduction to properly recognize all previous ground-state cooling demonstrations as well as recent progress in librational and multidimensional cooling (including 6 publications in the Nature family and 1 in Science).

We believe that our work, building on the advances reported in *Piotrowski et al. (2023)*, *Pontin et al. (2023)* and *Dania et al. (2025)*, still constitutes a significant step forward. In particular, the manuscript now highlights the following conceptual and technical novelties:

Two cavity modes for avoiding hybridization — For the first time, we demonstrate approaching the ground state of two mechanical degrees of freedom by separately coupling them to two orthogonally polarized cavity modes. As shown by *Toroš et al. (2021)*, using a single common cavity mode instead leads to unavoidable hybridization and imposes a stringent condition on the frequency difference between the translational modes (*Piotrowski et al., 2023*). In contrast, we are free to choose the same frequency for α and β . Crucially, our cavity eigenpolarizations are aligned with the particle axes, ensuring predominantly diagonal coupling and avoiding cross-couplings. Finally, we separate the detection of the two cavity modes via polarization, which for the first time allows us to directly identify and measure different librational modes in the cavity output.

Laying the groundwork for rotational quantum experiments — Cooling of two librational modes, which decouples them from the third librational degree of freedom (*Zielińska et al., 2024*), is a prerequisite for quantum experiments in rotational phase space, such as interference and revivals of rotational wave packets (*Ma et al., 2020*, *Stickler et al., 2018*, *2021*). These require control over at least two degrees of freedom, as a single cooled librational mode would inevitably thermalize under free evolution due to nonlinearities of the rotational motion (*Stickler et al., 2021*). This is a significant difference between the evolution in the rotational and in the translational phase space, where cooling of even a single linear degree of freedom has significant practical applications (*Bateman et al., 2014*).

Particle loading and characterization — Our particle loading mechanism based on laser-induced desorption operates in a novel regime with a launch substrate thickness about 100 times smaller and a laser pulse energy about 100 times lower than earlier work on laser-induced acoustic desorption in chemistry (*Zinovev et al., 2007*) and optomechanics (*Asenbaum et al., 2013*, *Bykov et al., 2019*, *Nikkhou et al., 2021*). This allows us to maintain a clean environment, significantly reducing the time from launching a nanorotor to ground-state cooling in high vacuum. Furthermore, we can characterize particle anisotropy in situ and, for the first time, achieve ground-state cooling of a variety of particles with different sizes and aspect ratios, all in the same setup on the same day. Therefore, testing new particles is significantly accelerated compared to previous experiments, which is essential for identifying particles with suitable geometries for simultaneous cooling of several degrees of freedom to very low temperatures.

Trap stability — We achieve two-dimensional (2D) cooling to the ground-state regime without having to strongly confine the third librational mode. Future rotational matter-wave interference experiments, which may require preparing and cooling large numbers of nearly identical nanorotors, benefit from the availability of reproducibly formed and stably trapped dumbbells. For such cylindrically symmetric particles, however, the librational

γ -motion is only weakly confined. This leads to new experimental challenges, for example, heating and instabilities due to mechanical resonances between the librational mode and the z -mode (*Pontin et al., 2023*). This dynamics has no equivalent in translational 2D cooling. Our experimental design enables us to cool two librations without strongly confining the third libration. This is in contrast to the approach followed by *Pontin et al. (2023)*, which relied on fully asymmetric nanoparticles.

Engineered cavity mode splitting — We introduce adjustable birefringence splitting between the orthogonally polarized cavity modes, ensuring that each librational frequency is aligned more closely with the corresponding cavity resonance it couples to. Tunable birefringence is of interest outside the levitated optomechanics community, for example, in the broader community of AMO physics, where engineering cavity birefringence is necessary to optimize light-matter interaction.

In the following, we provide a point-by-point reply to all comments, reproduced in full.

Referee 1

In “Quantum ground-state cooling of two librational modes of a nanorotor,” the authors report coherent-scattering-based cooling of two librational modes of an optically levitated dielectric particle to near the motional ground state (as low as $n = 0.2$ for a single mode, and as low as $n = 0.7$ and $n = 1.0$ simultaneously). They also describe their method for loading silica nanodimers and trimers into the cavity-enclosed optical tweezer trap, which enables demonstrably robust and selective cooling of different geometries.

Over the past three years, three similar results have been reported in Nature Physics. Earlier this year, Dania et. al. [46] reported coherent-scattering-based cooling of the librational mode of a levitated dielectric to an ultra-low occupation of $n = 0.04$. In 2023, Piotrowski et. al. [7] reported simultaneous ground cooling of two translational modes of a levitated dielectric to $n = 0.8$ and 0.8 ; and Pontin et. al. [45] reported simultaneous cooling of a 6 degrees of freedom (three librational, three translational) of a levitated particle.

The reported result is in many respects a synthesis of [45,46,7]. Like [46], a key enabling tool for ground state cooling is laser phase noise suppression (as emphasized in the recent News & Views article by Monteiro). Like [7], the main innovation (relative to [46]) is cooling of two degrees of freedom. Like [45], a key stated motivation is achieving improved orientational control of a nanoparticle.

Levitated optomechanics has attracted numerous outstanding research groups which have published important work before our own manuscript, which enabled us to go a significant step forward.

Cooling two librational modes is, however, more than a direct extension of one-dimensional (1D) librational or 2D translational cooling because it introduces constraints that do not have an analog in translational systems. *Pontin et al. (2023)* described already important details about the additional complexity involved in multi-dimensional cooling, such as mode hybridization and unstable trapping in high vacuum, with their particle still limited orders of magnitude distant from the quantum ground state.

The main novelty of the result appears to be its application to dimeric and trimeric particles, which the authors emphasize are examples of rotors (in [45,46,7], a single anisotropic particle was studied). This distinction—or reframing, if you consider an anisotropic particle to also be a rotor—allows the authors to discuss implications of multimode ground state cooling for rotational quantum experiments, in which the inherent nonlinearity of rotational states and their rich cross-couplings (e.g. to spin) offer new perspectives.

The key points of our paper are as now listed above:

- 2D librational cooling, as an essential starting point for future rotational revival experiments.
- proper characterization of particles, which is essential to understand and predict their dynamics.
- refined and cleaner particle loading mechanism, which speeds up the loading by an order of magnitude, already now.
- ground-state cooling of a series of different particle sizes and shapes, which solidly corroborates the universality of the scheme.

We put an emphasis on the characterization of the anisotropic nanorotors, but did not intend to differentiate between our particles from anisotropic particles in other experiments.

We use the name 'nanorotor' rather than 'anisotropic particle' (which nanorotors are), to emphasize the focus on the rotational physics, with the picture in mind that a rotor is defined by its dynamics, rather than its shape.

A strength of the article is its data presentation. Figures and plots are polished, visually pleasing, and described at a basic level appropriate for non-specialists—I have no reservations in this regard.

Thanks for your appreciation!

An (in my opinion significant) weakness is that, for non-specialist readers, none of the above context above is made clear. In fact, a naïve reading of the abstract and introduction might lead one to believe that the librational motion of a nanorotor has never been cooled to the ground state—[46] appears as an afterthought on the final line of the introduction. The same goes for multi-mode ground state cooling, whose precedence goes without mention ([7] is listed as one of 8 articles on the general topic of “quantum nanomechanics”). The manuscript contains many such omissions, as well as some questionable hyperbole and subjective language in places. As a specialist, this made me feel uncomfortable.

We have adapted the introduction and outlook to substantially widen the scope and references. The omissions have been corrected, and the language has been toned down to avoid any unintended hyperbole – which was mostly a result of joy and a little bit of pride after years of work – while retaining the key scientific messages.

Setting aside presentation strengths and weaknesses, whether the paper should be published in Nat. Phys. depends on how much weight one places on timeliness/framing versus scientific novelty. In the first regard, as outlined above, the result can be seen as a natural progression of a rapidly advancing field. In the latter, as a specialist, I don't think I learned anything new relative to [45,46,7]; however, I also appreciate the reframing of the problem around rotational dynamics, as well as the impressive diversity of ground state cooled rotors presented in Fig. 4—both of these I found novel relative to [45,46,7].

We agree that the question of importance is central and ultimately rests with the referees and the editor. Our view is that the present results are interesting and relevant because they open avenues in rotational state space that are significantly harder to access via translational degrees of freedom.

This has to do with the fact that rotational state space is closed and this shall facilitate coherence and interference experiments even when nanoparticles become so large that optical diffraction gratings or state expansion become challenging.

However, this is not possible with only a single librational degree of freedom under control but requires multidimensional librational cooling, as these degrees of freedom are not decoupled from each other – very much in contrast to the linear harmonic oscillators in a 3D optical trap.

Depending on the envisaged future interference experiments – revivals and rephasing of rotational wave packets, or free-fall nanoparticle interference – it may become important to load very many particles. Our experiment is the first to show the capability of launching, trapping, ground-state cooling and detecting multiple of these particles per day.

In addition, we think that our manuscript also provides small new insights for both theoreticians and experimentalists in the field, which are orthogonal to previously published work (for example, the librational mode coupling or our method of estimating the occupation

number for measurement series). We adapted the manuscript to better work out those details.

If the editor and other reviewers are similarly opinioned, and if the authors are willing to address the major concerns raised above, I would be inclined to recommend publication. Please see the comments below for specific recommendations. (Major and minor comments are interleaved.)

1. Abstract – “the reliable” = “reliable”, I believe
2. Abstract – “in a . . . cavity” = “into a . . . cavity”
3. Abstract – The first three sentences of the abstract would equally motivate [46], in which the librational mode of a nanorotor was cooled to $n = 0.04$. Consider tweaking the abstract to set these results apart. The abstract of [7] might be considered as an example.
4. p1c1p1 – “first examples of” = “first examples in”

Adapted.

5. p1c1p1 – “quantum rotors are only starting to receive the attention they deserve” – it’s unclear to me what the authors mean by “they deserve”. I suggest rewording with objective language here and in similar instances throughout the manuscript (see comments below).

Thanks for the hint. Adapted in several places.

6. p1c1p1-p1c2p1 – The omission of [46] in discussion of quantum rotors in all but the last sentence of the introduction (paragraphs 1-3 of the manuscript) strikes me as misleading; or at least, a misrepresentation of the recent history of the field. IMO this should be addressed. See also #8.

The introduction of our initial version followed a deliberate structure, starting with translational harmonic oscillators, then transitions to classical rotors, and finally to quantum rotors – first from a theoretical perspective and then experimentally. Within this narrative, the discussion of *Dania et al. (2025)* naturally appears in the section devoted to recent experimental progress on quantum rotors.

In revising the manuscript, we clarified this narrative. We also added a discussion that distinguishes our work from previous results, which makes the role of *Dania et al. (2025)* more prominent.

7. p1c1p2 – “without the need for any beam splitter or mirror and even for a” = “without the need for a beamsplitter or for a particle pinned in real space”
Here and through the manuscript, the use of intensifiers detracts from the potency of otherwise strong claims. I recommend adopting a neutral and factual tone, consistent with most published (perhaps not submitted) Nature manuscripts.

We acknowledge the reviewer’s perspective, but we put an intentional emphasize in this sentence. This phrase is a central motivation for exploring librational physics. We believe that this aspect (closed compact space) is going to be an essential element in future quantum experiments starting from levitated optomechanics. It is central to understanding why librations are “not just another degree of freedom” of many. We, however, followed the suggestion to remove “any”.

8. p1c2p2 – “Here, we demonstrate cooling of two librational degrees of freedom deep into their quantum states and simultaneously cool both modes such that the nanorotor’s orientation is defined at the quantum limit” – This transitional claim troubles me on three levels.

a. First, the preceding sentence announced that a nanorotor has been already been cooled to the ground state [46]—why then, should we now be interested in cooling two degrees of freedom? Since the previous paragraphs are about single mode ground state cooling (no mention of the results of [7] is made), this strikes me as a rhetorical slight of hand. The authors should motivate why they are pursuing a synthesis of [45,46,7]—or if they consider themselves distinct from [45,46,7], why they are distinct.

b. Second, in [46], the authors cool a single librational mode to $n = 0.04$ (96% purity). Here, the authors cool two modes to $n = 0.2$ (83%) and 0.5 (67%) separately, and $n = 0.8$ (56%) and 1.0 (50%) simultaneously. Consider removing the word “deep” or replacing it with more objective phrasing, to avoid diluting the potency of the claim.

c. I may be wrong, but it seems that the rotor is aligned, but not oriented, at the quantum limit, since one of the three librational modes remains uncooled. This is a key point in [46] that the authors appear to clarify in section III. Consider clarifying already here.

- a. Throughout the manuscript, we have reworked the citations and motivation for 2D librational cooling.
- b. There seems to be some ambiguity of terms in the community. To avoid this, we removed “deep”.
- c. Thank you for that catch. We changed it to “alignment” and briefly clarified it in the updated introduction.

9. p1c2p2 – “We achieve this by actively reducing laser phase noise by three orders of magnitude at both mechanical frequencies at both mechanical frequencies, and by exploiting two orthogonal, non-degenerate polarization modes of a high-finesse cavity” – The authors should give credit to [46,47] and [53], for the previous insights leading to this advance.

We removed the passage from the introduction entirely, but in the main text it is properly cited.

10. p1c2p2 – “we achieve unrivaled repetition rates [sic] enabling us to cool different nanorotors to the librational quantum ground state” It’s unclear in what sense/context the authors consider their repetition rates unrivaled, and why this enables ground state cooling. Consider rephrasing.

We agree that “unrivaled” is not a meaningful concept in this context. We stay with the simple numbers now.

11. p2c1p1 – “nominal diameter of 119 nm.” – can the authors truly resolve the diameter of their particles to better than 1%, and are the particles truly isotropic with that precision? This strikes me as remarkable. Consider adding a footnote or cross-reference to the methods, to explain.

This is the nominal mean diameter as specified by the manufacturer, with a standard deviation of 4 nm. In order to be consistent with other publications which mention the diameter as specified by the manufacturer, we keep 119 nm in the manuscript and clarified that it is the *mean* diameter.

12. p2c1p1 – (related to #11) Since the contents of this paragraph are key to the manuscript but require the (very nice) methods section to fully appreciate. I recommend citing the methods or cross-referencing section IV.

We now do so.

13. p2c1p3 – “Remarkably, the two librations couple selectively to orthogonal cavity polarizations. . .” The word ‘remarkable’ does not indicate if this selectively is physically important (it might just be “cool”, which it is), nor how it occurs. I recommend a neutral, factual rephrasing. If it is too complicated to package, then consider replacing “remarkably” with “notably” or “importantly”, and citing a previous paper and/or methods for elaboration.

We have toned down to simple fact stating. Again, sometimes we are surprised that things are as they are. But we agree that our own enthusiasm does not need to determine the language.

14. p2c1p3 - Eq. 2 and also Eq. 3 should have periods.

Done.

15. p2c1p4 and onward – Labels “ . . .” are interesting. I was resistant on first reading, but they’ve grown on me.

Thanks :-). Initially, we were also skeptical, but concluded that those would be the cleanest designation.

16. Fig. 1 caption – consider changing caption title from “infrared tweezer” to “optical tweezer”, to avoid confusing a general audience. (The first phrasing is cute in its concision, but irregular. Also, the word “infrared” is not defined or motivated in the caption.)

17. Fig. 1 caption – Consider changing “populates the optical cavity formed by two mirrors” = “populates an optical cavity formed by two mirrors oriented in the x-direction” for completeness.

18. Fig. 1 – the inset is not defined in the captions, and it not obvious what the horizontal axis offset frequency is. Consider specifying in either the caption or the figure.

Done.

19. p2c2p1 – “According to the theoretical coupling” I’m not sure what this phrase means/refers to. Consider rephrasing/clarifying.

We clarified the wording to:

”Based on the theoretical prediction, we identify the peak appearing in the orange (violet) power spectral density (PSD) of a_y (a_z) as α (β).”

20. p2c2p2 – p3c1p1 – “. . . the interaction Hamiltonian (2) predicts cooling via coherent scattering. . .”

Cite seminal work on coherent-scattering-based cooling here and elsewhere in the paragraph, where well-known facts are being repeated, for (I think appropriate) pedagogical reasons.

We now cite *Domokos et al. (2001)*, *Horak et al. (1997)*, *Vuletić and Chu (2000)*, and *Aspelmeyer et al. (2014)* throughout the paragraph.

21. p3c1p1 – “The librational oscillator is transferred from quantum number n to the next lower level $n-1$ ”
Since the oscillator is in a thermal state (not a number state), this phrasing is imprecise. I recommend revising here and below as appropriate, since this is canonical point.

Yes, we start from a thermal mixture. However, the transfer is still from state to state even in this thermal sum. For clarity, we reworded it to: “Each anti-Stokes scattering event reduces the oscillator quantum number by one, $n \rightarrow n - 1$, while increasing the photon energy from $\hbar\omega_l$ to $\hbar(\omega_l + \Omega_\mu)$. Starting from a thermal state, the net effect of these individual transitions is a decrease of the mean phonon occupation $\langle n \rangle$, i.e. cooling of the mechanical oscillator.”

22. p3c1p1 – Consider revising “. . . cools the mechanical motion” to “. . . cools the mechanical oscillator” and “heating the nanorotor motion” = “heating the nanorotor”, for clarity.

We followed the first recommendation, but believe that “heating the nanorotor motion” is more clear in discerning it from the internal temperature of the nanoparticle.

23. p3c1p1 – “On average, this imbalance transfers mechanical energy into photon energy, which exits the cavity” – this is nice, but it’s unclear if the thing that exits the cavity is the photon or energy. (minor squabble)

Reworded to:

”On average, the resulting imbalance transfers mechanical energy to the optical field, raising the energy of the photons that leave the cavity (*Delić et al., 2020, Horak et al., 1997*).”

24. p3c1p2 – “The equilibrium phonon occupation of the librational modes is therefore given by [Eq. 3]”
Since the paragraph does not contain enough information to derive Eq. 3, then a citation or a cross-reference to the methods should be given.

Thanks for the advice. We have added a reference to *Rabl et al. (2009)*.

25. p3c1p2 – “The phase noise contribution $n_\phi \approx S_\phi$. . . Is determined by laser phase noise S_ϕ . . .”
For clarity, I recommend defining n_ϕ explicitly rather than implicitly (it is not self-evident from Eq. 3); also S_ϕ needs to be identified as a spectral density.

Adapted.

26. p3c1p2 – “To mitigate it, we measure the phase noise using an unbalanced Mach-Zehnder interferometer”
Ref. [53] should be cited here. Also, the noise is not mitigated by measuring it, it’s mitigated by electrooptic feedback using the measurement as an error signal. This should be clarified.

The new statement reads:

”To reduce the effect of phase noise we measure it using an unbalanced Mach-Zehnder interferometer whose output is fed back into an electro-optic modulator (EOM) (*Dania et al., 2025, Parniak et al., 2021*).”

27. p3c1p2 – “pumping the chamber to about $2.5 - 3.5 \times 10^{-8}$ mbar” unless the pressure is known to this precision, I recommend changing to “about 3×10^{-8} mbar” with the implication that the pressure is known to one sig-fig (3 ± 1).

We adapted it to “ 3×10^{-8} mbar”.

28. p3c1p3 – “We fit the peaks with a Lorentzian profile to extract their frequencies, linewidths, and background noise levels” – I believe the authors also extract the area of the peaks (e.g. Fig. 2b). Also I believe its “fit the peaks to. . .”.

We determine the area of the peaks by summing up the data points of the PSDs, with background levels subtracted. We believe that this approach, in the presence of spectral artifacts, is more correct than just using the Lorentzian fit.

29. p3c1p4 – The discussion of sideband thermometry in this paragraph is canonical in the field of optomechanics and should be cited for general readers, IMO. E.g. Purdy et. al. PRA 2015 or (in the field of levitodynamics) Tebbenjohanns et. al. PRL 2020.

Our method of sideband thermometry differs from the sideband ratio method used so far in the field of optomechanics. We now refer to *Methods*, where we discuss both methods and properly cite *Purdy et al. (2015)* as well as describe our method in more detail.

30. p3c2p1 – Ditto.

We think that the conversion mean occupation number to ground-state population probability is textbook knowledge; we have added a paragraph to *Methods* instead of adding a citation from the field of optomechanics.

31. p3c2p1 – It’s not clear what counts as “good stability” in Fig. 2b. Since the gray data speaks for itself, consider removing “good” and replacing “seen when we plot” with “evidenced by plotting.” Or something thereabouts. (Minor squabble.)

The reviewer is right: it is always better to be quantitative than qualitative and evidence based than indicative. We have changed the sentence accordingly.

32. p3c2p1 – “Forces due to the intracavity light field act like a frequency tunable optical spring and damping” Cite canonical ref, and “damping” should be “damper” for grammatical correctness.

We corrected the sentence and added a reference to *Aspelmeyer et al. (2014)*.

33. p3c2p2 – “Although the frequency change is only on the level of a few per mille, we can clearly resolve it” Not clear what this sentence is supposed to convey. The stability of the system? (Is it uncommon to measure parts-per-thousand frequency shifts in optomechanics?). Also, what does “clearly resolved” mean? If this is just a qualitative aside, then consider saying so explicitly. Also, while I like “per mille”, note that it may distract readers who are used to “parts per thousand.”

We removed the sentence in question altogether.

34. p3c1p2 – “deep in the librational ground state” I recommend removing the word “deep” and, or, defining what it means (e.g. ground state purity, as discussed in [46]).

We removed it throughout the manuscript.

35. p4c1p1 – Consider changing “corresponding to a temperature as low as $T = \dots$ ” to “corresponding to an effective temperature of $T = \hbar\omega n/k_B = \dots$ ” since effective [modal] temperature has not been defined, and it is normally distinguished from [bulk] temperature.

This approximation holds only for the classical limit. We have added the precise equation following from the Bose-Einstein statistics to Methods.

36. p4c2p1 – “Such cooling at the quantum limit. . .” Consider placing this nice discussion in its own paragraph following the paragraph describing evidence for 2D ground-state cooling.

We followed the recommendations.

37. p5c1p1 – I like section IV and Fig. 4. They set the paper apart from [46,45,7], in my mind, and motivate the discussion of rotors. If push comes to shove (including with this referee) the distinction could be made more explicit.

We have added more detail to this section, which is indeed also important to us. These changes are now marked in blue in the adapted manuscript. Rotor physics lives from the fact that we can load sufficiently many different rotors and that we can characterize them. There is always more to control than for spherical bodies. And this is also what takes all the effort to get it done well.

38. p5c1p3 – The first paragraph of the conclusion is written in a different voice than the rest of the main text. Consider returning to active voice with personal pronouns (we), for consistency/clarity.

Thanks for pointing out this. We now make sure that the article maintains an active style and voice throughout the manuscript.

39. p5c1p3 – “cooling of two librational modes. . . can define its orientation a the quantum limit” – consider distinguishing “orientation” (requiring 3D cooling) from “alignment” (2D cooling), as discussed in section III.

This is an important point, which we now answer also in section III and we keep that definition now consistent throughout the text.

40. p5c2p2 – “Observing rotational matter-wave interference. . .” – Consider adding citation.

41. p6c1p1 – “Besides [interest] in fundamental quantum physics that can test macroscopicity” – ditto.

We now cite *Stickler et al. (2018)*, as well as *Schrinski et al. (2019)*.

42. p6c1p1 – Consider tying discussion of levitated viruses in the quantum regime back to early proposals by Tongcan Li and others (e.g. Yin & Li *Contemporary Physics*, 2017)

During compacting the conclusion, we removed the quantum aspects of levitated viruses.

Referee 2

The paper “Quantum ground-state cooling of two librational modes of a nanorotor” reports the simultaneous quantum ground-state cooling of two librational modes of a levitated silicon nanorotor held in an optical dipole trap. This work extends the paradigm of levitated optomechanics—traditionally focused on the center-of-mass motion of spherical nanoparticles—to rotational degrees of freedom inherent to nonspherical objects. Such systems possess rich internal dynamics, including torsional and librational modes, that can couple to optical and mechanical fields in ways unavailable to spheres. Demonstrating quantum control over these internal degrees of freedom represents a major step toward exploring the quantum mechanics of rotation and orientation. It opens perspectives for fundamental studies of macroscopic quantum phenomena, precision torque sensing, and the realization of hybrid quantum systems exploiting quantized rotational motion.

The authors present a well-designed experimental setup in which an anisotropic silicon nanorotor is optically trapped and coupled to a high-finesse cavity field. They map the system dynamics onto a standard optomechanical model, treating the two librational modes as harmonic oscillators interacting with the cavity via radiation-pressure-like coupling. The paper first demonstrates resolved-sideband cooling of a single librational mode to near its quantum ground state, followed by simultaneous cooling of two orthogonal modes, supported by calibrated noise spectra and thermometry analysis. The experimental protocol includes an automated trap-loading and feedback-cooling sequence enabling stable operation over a 28-hour run, attesting to the robustness of the setup. I appreciate both the clarity of presentation and the thoroughness of the data analysis. The achieved quantum control of two rotational modes constitutes an important milestone for optomechanics with nanorotors. Whether the work merits publication in *Nature Physics* will, however, depend on the resolution of several specific issues discussed below.

We thank the reviewer for their appreciation of our work :-).

Major points

1. Scope of the advance

Cooling of one librational mode has already been demonstrated by Dania et al. [Ref. 40]. The present work extends this to two dimensions. In how far does this represent a substantive conceptual advance rather than a technical extension? What specific difficulties prevented earlier experiments from reaching this regime, and what concrete obstacles were overcome here? This is a critical aspect—the answer must convincingly establish the significance of the advance.

This has been a legitimate question for all experiments after the first demonstration of 1D ground-state cooling in levitated optomechanics. We believe that every previous step was significant and important and that our present work represents a further significant advance in this progression.

In the present case, the key difficulty is achieving a regime in which two librational frequencies lie within a narrow cavity linewidth while maintaining linear tweezer polarization and low phase noise. Meeting it required several nontrivial developments:

- loading symmetric nanodumbbells whose α - and β -libration differs by only a few dozen kHz;
- designing the experiment to remain stable despite the presence of a largely free γ -mode, which would otherwise lead to parametric instabilities;
- (librational) modes that do not hybridize; and

- suppressing laser phase noise at *multiple* mechanical frequencies, which had not previously been necessary in levitated optomechanics.

2. Motivation

The motivation in the abstract and introduction appears rather weak. What exactly are the “plethora of phenomena and applications” the authors refer to? This should be stated more concretely. Moreover, the phrase “. . . are only starting to receive the attention they deserve” sounds somewhat defiant and should be reformulated in a more neutral scientific tone.

The abstract has been rewritten to include several suggestions from reviewers 1 and 2, including the presentation of the wider research field and references, the goals in the community, and the specific interest in rotational physics.

3. Mode identification (Section II)

In Section II, the authors present spectral densities and identify the observed peaks as corresponding to specific librational and rotational modes of the nanorotor. What is the exact procedure underlying this mode assignment? Is it purely based on comparison with *ab initio* theoretical modeling—using predicted mode frequencies and light-scattering selection rules—or are there independent experimental checks? How accurate are the theoretical frequency predictions, and how sensitive are they to uncertainties in particle geometry, trap alignment, or optical parameters? Clarifying how reliably peaks are assigned to particular modes is essential for evaluating the robustness of the analysis.

For particle identification, we solely rely on selection rules resulting from the interaction Hamiltonian. The theoretical model presented in *Methods* does not depend on the specific particle shape. The dipole approximation and the assumption of cavity eigenmodes polarized along y and z are sufficient to conclude that α (β) will only couple to the y (z) mode of the cavity.

We have improved the wording around the explanation. For completeness, we have also added a sentence regarding identification of the x , y , and z modes.

4. Experimental routine (Section IV)

Section IV describes an impressive 28-hour experimental run, but the experimental procedure is explained too briefly. A clearer account of the trapping, evacuation, and cooling sequence would be helpful.

We have revised the description of the experimental sequence from first LID loading to ground-state cooling in the methods section.

Smaller issues

Introduction

5) Explain why rotational wave functions can be “divided up and recombined naturally,” in contrast to translational wave functions.

We have adapted the sentence.

Librational trapped state can be described as a sum of rotational states, which are defined on a compact support. Although, once released, they can run out of phase, because angular momentum is fundamentally quantized, there is a smallest multiple of frequencies where they all get back in phase. This leads to a perfect recurrence of an arbitrary initial

state (*Stickler et al., 2018*). Therefore, observing quantum interference in rotational motion does not require a beam splitter or active intervention. This is in contrast to the linear translational phase space, where two wave functions of opposite momentum will separate from each other continuously until they receive another momentum kick.

6) Give a clearer account of the accessible mass range in nanorotor experiments. What is meant by a “small molecule”? The challenge appears to be reaching the MDa–GDa range, yet the conclusions suggest that smaller masses are needed to observe revivals. What is the optimal mass range for quantum experiments, and what limits it? This point needs to be clarified also in the conclusions.

We adapted the manuscript accordingly.

Rotational revivals have so far been seen with diatomic or polyatomic molecules (*Henrik Stapelfeldt (2003), Karamatskos et al. (2019)*), i.e. for objects many orders of magnitude smaller than the objects of our experiments. The revival time is given by $T = 2\pi I/\hbar$ and thus scales with the moment of inertia. For a dumbbell made of equal spheres, the moment of inertia is $I = (112/15)\pi\rho r^5$. The corresponding revival time is thus about 40 min for a GDa particle. The mass range is interesting, as it would allow operating in a regime in which gravitationally induced collapse models *Diosi (1984), Penrose (1996)* would already be testable. On the other hand, keeping coherence over this time is obviously very demanding, in particular as it is excluded to observe this in free fall on Earth (untrapped, no perturbing fields) over such a long time.

While there may be options to reduce the requirements and shorten the times, for example by going to fractional revival times, the most natural step forward towards genuine rotational quantum interference is to start from what has been achieved here and to reduce the size of the particle.

There are (at least) two reasons why this has not been done in the community yet: First, the general focus has been on pushing to higher masses—at the cost of increased difficulties in unambiguously observing genuine quantum effects. Second, coherent scattering cooling and the optical detection in feedback cooling depend on scattering, scaling with the particle polarizability and hence gets less efficient for smaller particles.

For 10 MDa masses, we see a sweet spot, where cooling into the quantum regime as well as realizing rotational revivals may still be feasible in the foreseeable future.

7) Clarify what is meant by “a state that is aligned at the quantum level” and “the nanorotor’s orientation is defined at the quantum limit.”

Our intention was to express that, with two librational modes cooled to $n \lesssim 1$, the uncertainty in alignment approaches its quantum zero-point width. We agree with the reviewer that our initial statement may be misunderstood.

We have therefore rephrased the statements to a state where the angular uncertainty is close to the quantum-mechanical zero-point fluctuations.

Section I

8) The optomechanical coupling constants g are not explained in the main text.

We have changed it to:

”The interaction between the particle and the cavity is described by the coupling constants

$g_\alpha \propto \chi_c - \chi_a$ and $g_\beta \propto \chi_c - \chi_b$ (see Eq. x) and the interaction potential ...”

9) The two librations couple selectively to the polarizations. While this is emphasized as remarkable, the authors do not explain it.

We replaced ”Remarkably, ...” with ”The interaction Hamiltonian reveals that the two librational degrees of freedom couple selectively to orthogonal cavity polarizations:”.

10) Clarify the phrase “According to the theoretical coupling.”

We clarified the wording to:

”Based on the theoretical prediction, we identify the peak appearing in the orange (violet) power spectral density (PSD) of a_y (a_z) as α (β).”

11) The explanation of cavity cooling seems overly complicated. What is meant by “At resonance, the cavity enhances the spectral mode density”?

We have simplified the sentence: ”At resonance, the interaction rate between the particle and the cavity light field increases.”

12) Before Eq. (3), what is Γ ? Should this be Γ_μ ? Perhaps remind the reader that $\mu = \alpha, \beta$, as must be inferred from Eq. (2).

We have revised the explanation and also implemented the advice.

13) The remarks on laser stabilization and background gas collisions could be moved to the appendix unless they are essential for understanding the data, which does not seem to be the case.

Stabilization of phase noise, as for *Dania et al. (2025)*, is one of the key requirements to achieve ground-state cooling. Upfront, it was not clear to us whether/how well the method developed in *Parniak et al. (2021)* can be extended to multiple frequencies.

From our perspective, it is important to mention the pressure during ground-state cooling in the main text. Since the end of section I, around Eq. (3), anyway lists the different heating mechanisms, we thought this would be a natural point to tie it in.

We therefore kept both remarks in the current version of the manuscript.

Section II

14) Specify which objects are cooled.

Done.

15) The notation A^\pm and $A_{S/aS}$ for scattering rates and PSD areas is somewhat confusing.

We removed $A_{S/aS}$ from the main text, as it was not necessary there.

Section IV

16) In Fig. 4b, the grey lines represent theoretical expectations. Does their thickness correspond to uncertainties?

The theoretically expected damping ratios were taken from *Ahn et al. (2018)* (Supplemental Material), which only included an uncertainty estimation for dumbbells.

For clarity, we removed the incomplete uncertainty estimate from the figure. As this only increases the apparent deviation between our data points and the theoretical prediction, we don't see a scientific issue with this omission.

The paper addresses a timely and important problem and presents technically demanding experiments with strong potential impact. The results are impressive, but the above points—particularly the justification of the conceptual advance (1) and the clarification of experimental and analytical details (3, 4)—should be convincingly addressed before the work can be considered for publication in *Nature Physics*.

We thank the reviewer for the careful and critical assessment. We agree that all points raised by both reviewers add to the required clarity. Because the question of conceptual advance has been asked by both reviewers, we have given the answer already above in the preface.

References

- Jonghoon Ahn, Zhujing Xu, Jaehoon Bang, Yu-Hao Deng, Thai M. Hoang, Qinkai Han, Ren-Min Ma, and Tongcang Li. Optically Levitated Nanodumbbell Torsion Balance and GHz Nanomechanical Rotor. *Phys. Rev. Lett.*, 121(3):033603, 2018. doi:10.1103/PhysRevLett.121.033603.
- Peter Asenbaum, Stefan Kuhn, Stefan Nimmrichter, Ugur Sezer, and Markus Arndt. Cavity cooling of free silicon nanoparticles in high vacuum. *Nat. Commun.*, 4(1):1–7, 2013. ISSN 2041-1723. doi:10.1038/ncomms3743.
- Markus Aspelmeyer, Tobias J. Kippenberg, and Florian Marquardt. Cavity optomechanics. *Rev. Mod. Phys.*, 86:1391–1452, Dec 2014. doi:10.1103/RevModPhys.86.1391.
- James Bateman, Stefan Nimmrichter, Klaus Hornberger, and Hendrik Ulbricht. Near-field interferometry of a free-falling nanoparticle from a point-like source. *Nature Communications*, 5(1):4788, 2014. ISSN 2041-1723. doi:10.1038/ncomms5788.
- Dmitry S. Bykov, Pau Mestres, Lorenzo Dania, Lisa Schmöger, and Tracy E. Northup. Direct loading of nanoparticles under high vacuum into a Paul trap for levitodynamical experiments. *Appl. Phys. Lett.*, 115(3):034101, 2019. ISSN 0003-6951. doi:10.1063/1.5109645.
- Lorenzo Dania, Oscar Schmitt Kremer, Johannes Piotrowski, Davide Candoli, Jayadev Vijayan, Oriol Romero-Isart, Carlos Gonzalez-Ballester, Lukas Novotny, and Martin Frimmer. High-purity quantum optomechanics at room temperature. *Nature Physics*, 21(10):1603–1608, 2025. ISSN 1745-2481. doi:10.1038/s41567-025-02976-9.
- Uroš Delić, Manuel Reisenbauer, Kahan Dare, David Grass, Vladan Vuletić, Nikolai Kiesel, and Markus Aspelmeyer. Cooling of a levitated nanoparticle to the motional quantum ground state. *Science*, 367:892–895, 2020. ISSN 0036-8075, 1095-9203. doi:10.1126/science.aba3993.
- Lajos Diosi. Gravitation and quantum-mechanical localization of macro-objects. *Phys. Lett.*, 105 A:199–202, 1984.
- Peter Domokos, Peter Horak, and Helmut Ritsch. Semiclassical theory of cavity-assisted atom cooling. *Journal of Physics B: Atomic, Molecular and Optical Physics*, 34(2):187, jan 2001. doi:10.1088/0953-4075/34/2/306.
- Tamar Seideman Henrik Stapelfeldt. Colloquium: Aligning molecules with strong laser pulses. *Rev. Mod. Phys.*, 75:543–557, 2003.
- Peter Horak, Gerald Hechenblaikner, Klaus M. Gheri, Herwig Stecher, and Helmut Ritsch. Cavity-induced atom cooling in the strong coupling regime. *Phys. Rev. Lett.*, 79:4974–4977, Dec 1997. doi:10.1103/PhysRevLett.79.4974.
- E. T. Karamatskos, S. Raabe, T. Mullins, A. Trabattoni, P. Stammer, G. Goldsztejn, R. R. Johansen, K. Dlugolecki, H. Stapelfeldt, M. J. J. Vrakking, S. Trippel, A. Rouzee, and J. Küpper. Molecular movie of ultrafast coherent rotational dynamics of ocs. *Nat. Commun.*, 10:3364, 2019. doi:10.1038/s41467-019-11122-y.
- Yue Ma, Kiran E. Khosla, Benjamin A. Stickler, and M. S. Kim. Quantum persistent tennis racket dynamics of nanorotors. *Phys. Rev. Lett.*, 125(5):053604, 2020. ISSN 0031-9007, 1079-7114. doi:10.1103/PhysRevLett.125.053604.

- M. Nikkhou, Y. H. Hu, J. A. Sabin, and J. Millen. Direct and clean loading of nanoparticles into optical traps at millibar pressures. *Photonics*, 8:9, 2021. doi:10.3390/photonics8110458.
- Michał Parniak, Ivan Galinskiy, Timo Zwettler, and Eugene S. Polzik. High-frequency broadband laser phase noise cancellation using a delay line. *Opt. Expr.*, 29(5):6935–6946, 2021. ISSN 1094-4087. doi:10.1364/OE.415942.
- R. Penrose. On gravity’s role in quantum state reduction. *Gen. Rel. Grav.*, 28:581–600, 1996. doi:10.1007/BF02105068.
- Johannes Piotrowski, Dominik Windey, Jayadev Vijayan, Carlos Gonzalez-Ballester, Andrés de los Ríos Sommer, Nadine Meyer, Romain Quidant, Oriol Romero-Isart, René Reimann, and Lukas Novotny. Simultaneous ground-state cooling of two mechanical modes of a levitated nanoparticle. *Nat. Phys.*, 19:1009–1013, 2023. ISSN 1745-2481. doi:10.1038/s41567-023-01956-1.
- A. Pontin, H. Fu, M. Toroš, T. S. Monteiro, and P. F. Barker. Simultaneous cavity cooling of all six degrees of freedom of a levitated nanoparticle. *Nat. Phys.*, 19(7):1003–1008, 2023. ISSN 1745-2473, 1745-2481. doi:10.1038/s41567-023-02006-6.
- T. P. Purdy, P.-L. Yu, N. S. Kampel, R. W. Peterson, K. Cicak, R. W. Simmonds, and C. A. Regal. Optomechanical raman-ratio thermometry. *Phys. Rev. A*, 92:031802, Sep 2015. doi:10.1103/PhysRevA.92.031802.
- P. Rabl, C. Genes, K. Hammerer, and M. Aspelmeyer. Phase-noise induced limitations on cooling and coherent evolution in optomechanical systems. *Phys. Rev. A*, 80(6):063819, 2009. ISSN 1050-2947, 1094-1622. doi:10.1103/PhysRevA.80.063819.
- Björn Schirnski, Stefan Nimmrichter, Benjamin A. Stickler, and Klaus Hornberger. Macroscopicity of quantum mechanical superposition tests via hypothesis falsification. *Phys. Rev. A*, 100:032111, Sep 2019. doi:10.1103/PhysRevA.100.032111.
- Benjamin A. Stickler, Birthe Papendell, Stefan Kuhn, Björn Schirnski, James Millen, Markus Arndt, and Klaus Hornberger. Probing macroscopic quantum superpositions with nanorotors. *New J. Phys.*, 20(12):122001, 2018. ISSN 1367-2630. doi:10.1088/1367-2630/aaee4.
- Benjamin A. Stickler, Klaus Hornberger, and M. S. Kim. Quantum rotations of nanoparticles. *Nat. Rev. Phys.*, 3(8):589–597, 2021. ISSN 2522-5820. doi:10.1038/s42254-021-00335-0.
- Marko Toroš, Uroš Deli ć, Fagin Hales, and Tania S. Monteiro. Coherent-scattering two-dimensional cooling in levitated cavity optomechanics. *Phys. Rev. Res.*, 3:023071, Apr 2021. doi:10.1103/PhysRevResearch.3.023071.
- Vladan Vuletić and Steven Chu. Laser cooling of atoms, ions, or molecules by coherent scattering. *Phys. Rev. Lett.*, 84:3787–3790, Apr 2000. doi:10.1103/PhysRevLett.84.3787.
- J. A. Zielińska, F. van der Laan, A. Norrman, R. Reimann, M. Frimmer, and L. Novotny. Long-axis spinning of an optically levitated particle: A levitated spinning top. *Phys. Rev. Lett.*, 132:253601, Jun 2024. doi:10.1103/PhysRevLett.132.253601.
- A. V. Zinovev, I. V. Veryovkin, J. F. Moore, and M. J. Pellin. Laser-driven acoustic desorption of organic molecules from back-irradiated solid foils. *Analytical Chemistry*, 79:8232–8241, 2007. doi:10.1021/ac070584o.

Reply to the Reviewers

We thank both referees for their time and their positive feedback.

In the following, we give a point-by-point reply to all comments, which we reproduce in full.

Referee 2

I have carefully read both referee reports, the authors' response, and the revised manuscript. Overall, I am satisfied with the authors' replies to the questions and comments raised in the first round, and with the corresponding changes made to the manuscript. In both reports, the most critical issue concerned the novelty of the present work in comparison to previous studies. While the extension from cooling a single librational mode to two modes still appears, at first sight, somewhat incremental, the authors convincingly argue that this step is essential for accessing and exploiting genuine nanorotor physics, which naturally lives in a two-dimensional phase space.

In the revised manuscript, this motivation is presented more clearly, and I largely follow and agree with the authors' line of argumentation. The comparison with previous literature has been improved, and both the style and tone of the presentation are more balanced and precise.

We appreciate the comments.

One point, however, still merits attention. I was surprised by the paragraph starting with "Here, we demonstrate . . ." on page 1. This statement is noticeably weaker than the case made in the authors' response to the referees and reflects only part of the advances discussed there. In particular, progress regarding particle loading and characterization, as well as improvements in trap stability, are not mentioned at all. If the article is intended to demonstrate key enabling steps towards nanorotor optomechanics, I would recommend that all of these advances be clearly reflected already in the introduction, so that the scope and significance of the work are immediately apparent to the reader. This remark is meant as an optional suggestion for further strengthening the presentation.

We have revised the introduction to highlight:

- the usage of two cavity modes for avoiding hybridization and direct detection of the modes
- the engineered cavity mode splitting
- the extension of the phase noise reduction to multiple frequencies
- our advances in laser-induced desorption and the reduction of the laser pulse energy
- the multiple loading and characterization of the particles

Overall, I am satisfied with the revised manuscript and recommend acceptance.

We thank the referee for this positive assessment.

Referee 1

The authors have done a commendable job of responding to the comments of both referees, including major revisions to the manuscript that address most of my minor concerns and some of my major concerns. I appreciate their efforts.

Thanks again.

In my original review, I supported publication contingent on addressal of one major concern seemingly shared by Referee #2, namely: The abstract and the introduction are written in a way that underplays the fact that the librational mode of a nanorotor has already been ground state cooled. So as not to mislead general readers (specialists in levitodynamics will read between the lines), I suggested that the authors make it clear from the outset that *Dania et al.* have already achieved ground state cooling of a nanorotor, and to explicitly state (1) why and (2) how they have moved beyond this achievement, to cooling of two librational modes simultaneously.

The authors have given convincing answers to (1) and (2) in their rebuttal; however, the revised abstract and introduction still miss the mark, in my opinion. The achievement of *Dania et al.* is still not mentioned until the end of the paragraph #3, appearing as an afterthought. Moreover, the innovations necessary to achieve 2D librational mode cooling—multimode phase-noise eating, loading highly asymmetric particles, dual-mode cavity optomechanics with immunity to mode hybridization—while partially incorporated into the introduction, are not explicitly contextualized with the achievements of *Dania et al.* or (equally relevant) *Pontin et al.*

We have rewritten the first part of the introduction to follow the suggestion of both referees in the previous and current rounds of reviews. In particular *Dania et al.* is now featured in the first paragraph.

Additionally, before describing our own work we emphasize all progress in librational cooling, also referring to *Pontin et al.* and librational ground state cooling by *Dania et al.* stating that this development is "culminating in the cooling to a high-purity quantum ground state for a single librational mode" *Dania et al.*

As suggested by the referee, the subsequent paragraph then builds thematically on this result (historically, our work has developed independently), including the main experimental advances that were required to progress from 1D to 2D cooling and from a single type of particles to several different per day.

The referee suggested that the abstract might also be a suitable place for reference to *Dania et al.*. Because our work builds on several prior achievements, from the underlying theoretical idea of coherent-scattering cooling to advances in linear motion, particle-loading and learnings from other rotational experiments, the abstract cannot do justice to all relevant prior work. We therefore cite these contributions explicitly and extensively in the introduction, with *Dania et al.* singled out in the first and 5th paragraph.

That said, I appreciate that the author's focus on rotational dynamics is a framing choice that has its merits. *Dania et. al.* made little mention of rotational dynamics in its introduction, and was concerned more with the fact that they had achieved record ground state cooling from room temperature. *Pontin et. al.*, on the other hand, was focused on the mechanism and motivation for cooling all 6 translational and rotational degrees of freedom of a nanoparticle, rather than the specific challenges of and outlook for rotational ground state cooling. By focusing on the latter, the authors can lay claim to "giving nanorotors the attention they deserve." In my mind they miss the opportunity to *fully* right the ship and present a historically balanced point of view of where the field of levitodynamics is going, but that doesn't demerit their results.

We are grateful to take the opportunity suggested by the referee. We have expanded the section about previous work on rotational levitodynamics, including new references, now starting with *Arita et al.* which shows that this research programme has been pursued in the community for more than a dozen years.

Bottom line, while I would have framed the abstract/intro differently, I think the paper is timely, scientifically sound, well-presented overall, and deserving of publication in Nat. Physics.

I nevertheless would consider the following suggestions for clarity.

1. The abstract and the introduction suggest that cooling to two-dimensional librational ground state is a prerequisite for various fundamental experiments. As stated, it's not clear if this is a *necessary* or *sufficient* condition. Consider clarifying this in the intro (e.g. par #3).

Two-mode librational cooling is the minimal requirement to define the rotor axis in 3D, enabling reproducible release into near-free rotational evolution and preserving coherence for subsequent rotational interferometry.

We write in the introduction that two-dimensional cooling is a necessary condition for starting orientational revival experiments. It is not yet a sufficient condition as the moment of inertia has to be chosen such that the revival times become realistic. This is possible by moving to somewhat smaller masses, as now better explained in the outlook.

2. (line 15) It's not clear what "nearly ideal realization of such oscillators" means. Consider replacing "such" with something specific.

We have changed it to "Optically levitated nanoparticles provide a realization particularly close to the ideal harmonic oscillator: [...]". Why the realization of the oscillator is nearly ideal is explained in the [...] sentence that follows.

3. (line 21) "Complementary to these achievements" appears to refer to ground state cooling of multiple center-of-mass (COM) degrees of freedom; however, one must read carefully to catch this distinction, since the first paragraph is about levitated particles in general (not specifically COM), until the last sentence. Since it's crucial to the narrative, consider clarifying.

Because *Dania et. al.* is now a reference embedded into the front of the introduction, on par with the highlights in translation, the paragraph has been adapted and should be clear.

4. (line 66) *Dania et. al.* is introduced here at the end of an extended discussion about center-of-mass-mode levitodynamics and rotational levitodynamics. The authors say this placement was based on “a deliberate structure, starting with translational harmonic oscillators, then transitions to classical rotors, and finally to quantum rotors – first from a theoretical perspective and then experimentally”. This may be the case, but since the novelty of the paper is *multimode* quantum rotors, then the deliberate structure is missing a critical logical step. I don’t have an easy solution to this issue, but encourage the authors again to consider it.

We added *Dania et al.* to the first paragraph of the introduction as mentioned above

We have also additionally strengthened the motivation for rotational physics, included more citations and explained in more detail the need for two-dimensional ground-state cooling before explaining the step from 1D to 2D, which is the direct link from *Dania et al.* to us.

5. (line 67+) This paragraph still suffers from issues raised in my comment #8a, and Referee #2’s “major” comments 1-2: the authors announce that have achieve 2D librational ground state cooling without explicitly motivating it or explaining how they’ve achieved it. They’ve inserted the statement that 2D librational ground state cooling is central to “studies of quantum tennis racket flips,” but that seems like a rather narrow motivation (echoing referee #2). They’ve also highlighted that librational modes avoid optomechanical hybridization and that they separately address two cavity modes; however, they don’t explicitly connect this to challenge of 2D cooling. Finally, they don’t mention (as they emphasize in in their referee rebuttal) that 2D cooling is also made possible by multimode phase noise eating and the use of highly anisotropic particles relative to *Dania et. al.*. Some minor revision could at least partially resolve these issues.

In the revision the recommendations of both referees are included. The introduction has grown in length and content, accordingly. We feel that the present text now covers history, experiment and outlook in a good balance, also the connection from the prior work *Dania et. al.* to the additional challenges in our work.

6. (line 135) “. . . we split the transmitted cavity light into its eigenmodes. . .” Since this is a description of experimental method, the authors should describe *how* they split the polarizations. (Maybe I missed it?)

We have changed the text to “[. . .] we split the transmitted cavity light via a polarizing beam splitter into the cavity’s eigenmodes [. . .]”.

7. (line 186+) “To reduce the effect of phase noise, we measure it using a MZI whose output is fed back into an EOM. In this way. . .” As mentioned in my first review, phase noise is reduced by feedback, not by measurement. A native English speaker who reads this sentence will think it’s the latter, which is incorrect. Since this is a key point, I recommend fixing the grammar.

Thanks for spotting this. We have modified the sentence to: “To reduce the effect of phase noise we actively stabilize the laser phase using feedback derived from an unbalanced Mach-Zehnder interferometer [. . .]”.

8. (line 239) “we determine the SDEV of the rotational mode as . . . corresponding to an effective temperature as low as $T = 28 \pm 2 \mu\text{K}$.” Two comments: (1) I believe that the authors mean “motion” (more precisely, librational amplitude), not “mode”; (2) if “as low as $T = 28 \mu\text{K}$ ” is meant as a cross-check of the sideband thermometry measurement, it begs the question whether $28 \mu\text{K}$ is consistent with $n = 0.21$ on line 213. Is it? Consider stating in the text.

We have changed “standard deviation of the rotational mode” to “standard deviation of the librational amplitude”.

The temperature is calculated from the occupation number. To state this more clearly, we have changed the wording in *Methods* to “Knowing n , we estimate the mode temperature T by assuming the Bose-Einstein distribution for a quantum harmonic oscillator in thermal equilibrium.”

The next few comments concern the conclusion, which is still difficult to understand as written.

9. (line 355) “We have demonstrated ground state cooling of both rotational DOFs individually. . . .” Both rotational DOFs of what? Please state explicitly what was accomplished.

10. (line 359) “we align the nanorotor. . . .” Which nanorotor? Please state explicitly.

This is an important point, we have streamlined the conclusion and are now also more precise in summarizing what has been achieved in which system and how we are planning to proceed to quantum experiments.

11. (line 362) “Combined with the capability for fast and repeatable loading and cooling. . . . these results provide. . . .” It is unclear if this is a general statement, or if the authors are stating that “fast and repeatable loading. . . .” is one of their accomplishments. Please be explicit.

We have followed the suggestion and now write: “Combined with *our* capability [. . .]”

12. (line 366) “With a mass around 4 GDa. . . .” This sentence (and therefore paragraph) seems to come out of nowhere. The unit “GDa” is not previously used or defined in the text (it is implicitly defined later in the sentence by equating 10 MDa with the mass of two 20 nm silica spheres), nor have the authors stated in the conclusion or elsewhere in the whether the spectrum of devices studied includes a 4 GDa rotor. Please be explicit to help guide the reader.

We now clarify that the particle mass reported in our 2D cooling experiment was approximately 4×10^9 u. We also avoid Dalton (Da) as a unit and stay with atomic mass units (u).

13. (line 381-382) “In addition, quantum nanorotors in the mass range of 0.1 – 1 MDa” Ditto. How does this relate to the present work? Please make the connection explicit.

14. (line 385) “The mass scale of 10 MDa. . . .” Ditto.

We have adapted the final section to relate the mass scale of tobacco mosaic viruses to dumbbells formed by 20 nm nanospheres. And we now stay consistently in this mass range also for a predicted resonant torque sensitivity.

In “Quantum ground-state cooling of two librational modes of a nanorotor,” the authors report coherent-scattering-based cooling of two librational modes of an optically levitated dielectric particle to near the motional ground state (as low as $n = 0.2$ for a single mode, and as low as $n = 0.7$ and $n = 1.0$ simultaneously). They also describe their method for loading silica nanodimers and trimers into the cavity-enclosed optical tweezer trap, which enables demonstrably robust and selective cooling of different geometries.

Over the past three years, three similar results have been reported in Nature Physics. Earlier this year, Dania et. al. [46] reported coherent-scattering-based cooling of the librational mode of a levitated dielectric to an ultralow occupation of $n = 0.04$. In 2023, Piotrowski et. al. [7] reported simultaneous ground cooling of two *translational* modes of a levitated dielectric to $n = 0.8$ and 0.8 ; and Pontin et. al. [45] reported simultaneous cooling of a 6 degrees of freedom (three librational, three translational) of a levitated particle.

The reported result is in many respects a synthesis of [45,46,7]. Like [46], a key enabling tool for ground state cooling is laser phase noise suppression (as emphasized in the recent News & Views article by Monteiro). Like [7], the main innovation (relative to [46]) is cooling of two degrees of freedom. Like [45], a key stated motivation is achieving improved orientational control of a nanoparticle.

The main novelty of the result appears to be its application to dimeric and trimeric particles, which the authors emphasize are examples of rotors (in [45,46,7], a single anisotropic particle was studied). This distinction---or reframing, if you consider an anisotropic particle to also be a rotor---allows the authors to discuss implications of multimode ground state cooling for *rotational* quantum experiments, in which the inherent nonlinearity of rotational states and their rich cross-couplings (e.g. to spin) offer new perspectives.

A strength of the article is its data presentation. Figures and plots are polished, visually pleasing, and described at a basic level appropriate for non-specialists---I have no reservations in this regard.

An (in my opinion significant) weakness is that, for non-specialist readers, *none of the above context above is made clear*. In fact, a naïve reading of the abstract and introduction might lead one to believe that the librational motion of a nanorotor has never been cooled to the ground state---[46] appears as an afterthought on the final line of the introduction. The same goes for multi-mode ground state cooling, whose precedence goes without mention ([7] is listed as one of 8 articles on the general topic of “quantum nanomechanics”). The manuscript contains many such omissions, as well as some questionable hyperbole and subjective language in places. As a specialist, this made me feel uncomfortable.

Setting aside presentation strengths and weaknesses, whether the paper should be published in Nat. Phys. depends on how much weight one places on timeliness/framing versus scientific novelty. In the first regard, as outlined above, the result can be seen as a natural progression of a rapidly advancing field. In the latter, as a specialist, I don’t think I learned anything new relative to [45,46,7]; however, I also appreciate the reframing of the problem around rotational dynamics, as well as the impressive diversity of ground state cooled rotors presented in Fig. 4—both of these I found novel relative to [45,46,7].

If the editor and other reviewers are similarly opinioned, and if the authors are willing to address the major concerns raised above, I would be inclined to recommend publication. Please see the comments below for specific recommendations. (Major and minor comments are interleaved.)

(pXcYpZ = page X, column Y, paragraph Z)

1. Abstract – “the reliable” = “reliable”, I believe
2. Abstract – “in a ... cavity” = “into a... cavity”
3. Abstract – The first three sentences of the abstract would equally motivate [46], in which the librational mode of a nanorotor was cooled to $n = 0.04$. Consider tweaking the abstract to set these results apart. The abstract of [7] might be considered as an example.
4. p1c1p1 – “first examples of” = “first examples in”
5. p1c1p1 – “quantum rotors are only starting to receive the attention they deserve” – it’s unclear to me what the authors mean by “they deserve”. I suggest rewording with objective language here and in similar instances throughout the manuscript (see comments below).
6. p1c1p1-p1c2p1 – The omission of [46] in discussion of quantum rotors in all but the last sentence of the introduction (paragraphs 1-3 of the manuscript) strikes me as misleading; or at least, a misrepresentation of the recent history of the field. IMO this should be addressed. See also #8.
7. p1c1p2 – “without the need for any beam splitter or mirror and even for a” = “without the need for a beamsplitter or for a particle pinned in real space” Here and through the manuscript, the use of intensifiers detracts from the potency of otherwise strong claims. I recommend adopting a neutral and factual tone, consistent with most published (perhaps not *submitted*) Nature manuscripts.
8. p1c2p2 – “Here, we demonstrate cooling of two librational degrees of freedom deep into their quantum states and simultaneously cool both modes such that the nanorotor’s orientation is defined at the quantum limit” – This transitional claim troubles me on three levels.
 - a. First, the preceding sentence announced that a nanorotor has been already been cooled to the ground state [46]—why then, should we now be interested in cooling two degrees of freedom? Since the previous paragraphs are about single mode ground state cooling (no mention of the results of [7] is made), this strikes me as a rhetorical slight of hand. The authors should motivate why they are pursuing a synthesis of [45,46,7]---or if they consider themselves distinct from [45,46,7], why they are distinct.
 - b. Second, in [46], the authors cool a single librational mode to $n = 0.04$ (96% purity). Here, the authors cool two modes to $n = 0.2$ (83%) and 0.5 (67%) separately, and $n = 0.8$ (56%) and 1.0 (50%) simultaneously. Consider removing the word “deep” or replacing it with more objective phrasing, to avoid diluting the potency of the claim.
 - c. I may be wrong, but it seems that the rotor is *aligned*, but not *oriented*, at the quantum limit, since one of the three librational modes remains uncooled. This is a key point in [46] that the authors appear to clarify in section III. Consider clarifying already here.
9. p1c2p2 – “We achieve this by actively reducing laser phase noise by three orders of magnitude at both mechanical frequencies at both mechanical frequencies, and by exploiting two orthogonal, non-degenerate polarization modes of a high-finesse cavity” – The authors should give credit to [46,47] and [53], for the previous insights leading to this advance.
10. p1c2p2 – “we achieve unrivaled repetition rates [sic] enabling us to cool different nanorotors to the librational quantum ground state” It’s unclear in what sense/context the authors consider their repetition rates unrivaled, and why this enables ground state cooling. Consider rephrasing.
11. p2c1p1 – “nominal diameter of 119 nm..” – can the authors truly resolve the diameter of their particles to better than 1%, and are the particles truly isotropic with that precision? This strikes me as remarkable. Consider adding a footnote or cross-reference to the methods, to explain.

12. p2c1p1 – (related to #11) Since the contents of this this paragraph are key to the manuscript but require the (very nice) methods section to fully appreciate. I recommend citing the methods or cross-referencing section IV.
13. p2c1p3 – “Remarkably, the two librations couple selectively to orthogonal cavity polarizations...” The word ‘remarkable’ does not indicate if this selectively is physically important (it might just be “cool”, which it is), nor how it occurs. I recommend a neutral, factual rephrasing. If it is too complicated to package, then consider replacing “remarkably” with “notably” or “importantly”, and citing a previous paper and/or methods for elaboration.
14. p2c1p3 - Eq. 2 and also Eq. 3 should have periods.
15. p2c1p4 and onward – Labels ① ② ③ are interesting. I was resistant on first reading, but they’ve grown on me.
16. Fig. 1 caption – consider changing caption title from “infrared tweezer” to “optical tweezer”, to avoid confusing a general audience. (The first phrasing is cute in its concision, but irregular. Also, the word “infrared” is not defined or motivated in the caption.)
17. Fig. 1 caption – Consider changing “populates the optical cavity formed by two mirrors” = “populates an optical cavity formed by two mirrors oriented in the x-direction” for completeness.
18. Fig. 1 – the inset is not defined in the captions, and it not obvious what the horizontal axis offset frequency is. Consider specifying in either the caption or the figure.
19. p2c2p1 – “According to the theoretical coupling” I’m not sure what this phrase means/refers to. Consider rephrasing/clarifying.
20. p2c2p2 – p3c1p1 – “...the interaction Hamiltonian (2) predicts cooling via coherent scattering...” Cite seminal work on coherent-scattering-based cooling here and elsewhere in the paragraph, where well-known facts are being repeated, for (I think appropriate) pedagogical reasons.
21. p3c1p1 – “The librational oscillator is transferred from quantum number n to the next lower level $n-1$ ” Since the oscillator is in a thermal state (not a number state), this phrasing is imprecise. I recommend revising here and below as appropriate, since this is canonical point.
22. p3c1p1 – Consider revising “...cools the mechanical motion” to “...cools the mechanical oscillator” and “heating the nanorotor motion” = “heating the nanorotor”, for clarity.
23. p3c1p1 – “On average, this imbalance transfers mechanical energy into photon energy, which exits the cavity” – this is nice, but it’s unclear if the thing that exits the cavity is the photon or energy. (minor squabble)
24. p3c1p2 – “The equilibrium phonon occupation of the librational modes is therefore given by [Eq. 3]” Since the paragraph does not contain enough information to derive Eq. 3, then a citation or a cross-reference to the methods should be given.
25. p3c1p2 – “The phase noise contribution $n_{\phi} \approx S_{\phi}$... Is determined by laser phase noise S_{ϕ} ...” For clarity, I recommend defining n_{ϕ} explicitly rather than implicitly (it is not self-evident from Eq. 3); also S_{ϕ} needs to be identified as a spectral density.
26. p3c1p2 – “To mitigate it, we measure the phase noise using an unbalanced Mach-Zehnder interferometer” Ref. [53] should be cited here. Also, the noise is not mitigated by measuring it, it’s mitigated by electrooptic feedback using the measurement as an error signal. This should be clarified.

27. p3c1p2 – “pumping the chamber to about $2.5 - 3.5 \times 10^{-8}$ mbar” unless the pressure is known to this precision, I recommend changing to “about 3×10^{-8} mbar” with the implication that the pressure is known to one sig-fig (3 ± 1).
28. p3c1p3 – “We fit the peaks with a Lorentzian profile to extract their frequencies, linewidths, and background noise levels” – I believe the authors also extract the area of the peaks (e.g. Fig. 2b). Also I believe its “fit the peaks to...”.
29. p3c1p4 – The discussion of sideband thermometry in this paragraph is canonical in the field of optomechanics and should be cited for general readers, IMO. E.g. Purdy et. al. PRA 2015 or (in the field of levitodynamics) Tebbenjohanns et. al. PRL 2020.
30. p3c2p1 – Ditto.
31. p3c2p1 – It’s not clear what counts as “good stability” in Fig. 2b. Since the gray data speaks for itself, consider removing “good” and replacing “seen when we plot” with “evidenced by plotting.” Or something thereabouts. (Minor squabble.)
32. p3c2p1 – “Forces due to the intracavity light field act like a frequency tunable optical spring and damping” Cite canonical ref, and “damping” should be “damper” for grammatical correctness.
33. p3c2p2 – “Although the frequency change is only on the level of a few per mille, we can clearly resolve it” Not clear what this sentence is supposed to convey. The stability of the system? (Is it uncommon to measure parts-per-thousand frequency shifts in optomechanics?). Also, what does “clearly resolved” mean? If this is just a qualitative aside, then consider saying so explicitly. Also, while I like “per mille”, note that it may distract readers who are used to “parts per thousand.”
34. p3c1p2 – “deep in the librational ground state” I recommend removing the word “deep” and, or, defining what it means (e.g. ground state purity, as discussed in [46]).
35. p4c1p1 – Consider changing “corresponding to a temperature as low as $T = \dots$ ” to “corresponding to an effective temperature of $T = \hbar \omega_n / k_B = \dots$ ” since effective [modal] temperature has not been defined, and it is normally distinguished from [bulk] temperature.
36. p4c2p1 – “Such cooling at the quantum limit...” Consider placing this nice discussion in its own paragraph following the paragraph describing evidence for 2D ground state cooling.
37. p5c1p1 – I like section IV and Fig. 4. They set the paper apart from [46,45,7], in my mind, and motivate the discussion of rotors. If push comes to shove (including with this referee) the distinction could be made more explicit.
38. p5c1p3 – The first paragraph of the conclusion is written in a different voice than the rest of the main text. Consider returning to active voice with personal pronouns (we), for consistency/clarity.
39. p5c1p3 – “cooling of two librational modes...can define its orientation at the quantum limit” -- consider distinguishing “orientation” (requiring 3D cooling) from “alignment” (2D cooling), as discussed in section III.
40. p5c2p2 – “Observing rotational matter-wave interference...” – Consider adding citation.
41. p6c1p1 – “Besides [interest] in fundamental quantum physics that can test macroscopicity” – ditto.
42. p6c1p1 – Consider tying discussion of levitated viruses in the quantum regime back to early proposals by Tongcan Li and others (e.g. Yin & Li Contemporary Physics, 2017)

The authors have done a commendable job of responding to the comments of both referees, including major revisions to the manuscript that address most of my minor concerns and some of my major concerns. I appreciate their efforts.

In my original review, I supported publication contingent on addressal of one major concern seemingly shared by Referee #2, namely: The abstract and the introduction are written in a way that underplays the fact that the librational mode of a nanorotor has already been ground state cooled. So as not to mislead general readers (specialists in levitodynamics will read between the lines), I suggested that the authors make it clear from the outset that *Dania et. al.* have already achieved ground state cooling of a nanorotor, and to explicitly state (1) why and (2) how they have moved beyond this achievement, to cooling of two librational modes simultaneously.

The authors have given convincing answers to (1) and (2) in their rebuttal; however, the revised abstract and introduction still miss the mark, in my opinion. The achievement of *Dania et. al.* is still not mentioned until the end of the paragraph #3, appearing as an afterthought. Moreover, the innovations necessary to achieve 2D librational mode cooling---multimode phase-noise eating, loading highly asymmetric particles, dual-mode cavity optomechanics with immunity to mode hybridization---while partially incorporated into the introduction, are not explicitly contextualized with the achievements of *Dania et. al.* or (equally relevant) *Pontin et. al.*

That said, I appreciate that the author's focus on rotational dynamics is a framing choice that has its merits. *Dania et. al.* made little mention of rotational dynamics in its introduction, and was concerned more with the fact that they had achieved record ground state cooling from room temperature. *Pontin et. al.*, on the other hand, was focused on the mechanism and motivation for cooling all 6 translational and rotational degrees of freedom of a nanoparticle, rather than the specific challenges of and outlook for rotational ground state cooling. By focusing on the latter, the authors can lay claim to "giving nanorotors the attention they deserve." In my mind they miss the opportunity to *fully* right the ship and present a historically balanced point of view of where the field of levitodynamics is going, but that doesn't demerit their results.

Bottom line, while I would have framed the abstract/intro differently, I think the paper is timely, scientifically sound, well-presented overall, and deserving of publication in Nat. Physics.

I nevertheless would consider the following suggestions for clarity.

1. The abstract and the introduction suggest that cooling to two-dimensional librational ground state is a prerequisite for various fundamental experiments. As stated, it's not clear if this is a *necessary* or *sufficient* condition. Consider clarifying this in the intro (e.g. par #3).
2. (line 15) It's not clear what "nearly ideal realization of such oscillators" means. Consider replacing "such" with something specific.
3. (line 21) "Complementary to these achievements" appears to refer to ground state cooling of multiple center-of-mass (COM) degrees of freedom; however, one must read carefully to

catch this distinction, since the first paragraph is about levitated particles in general (not specifically COM), until the last sentence. Since it's crucial to the narrative, consider clarifying.

4. (line 66) *Dania et. al.* is introduced here at the end of an extended discussion about center-of-mass-mode levitodynamics and rotational levitodynamics. The authors say this placement was based on “a deliberate structure, starting with translational harmonic oscillators, then transitions to classical rotors, and finally to quantum rotors - first from a theoretical perspective and then experimentally”. This may be the case, but since the novelty of the paper is *multi-mode* quantum rotors, then the deliberate structure is missing a critical logical step. I don't have an easy solution to this issue, but encourage the authors again to consider it.

5. (line 67+) This paragraph still suffers from issues raised in my comment #8a, and Referee #2's “major” comments 1-2: the authors announce that have achieve 2D librational ground state cooling without explicitly motivating it or explaining how they've achieved it. They've inserted the statement that 2D librational ground state cooling is central to “studies of quantum tennis racket flips,” but that seems like a rather narrow motivation (echoing referee #2). They've also highlighted that librational modes avoid optomechanical hybridization and that they separately address two cavity modes; however, they don't explicitly connect this to challenge of 2D cooling. Finally, they don't mention (as they emphasize in in their referee rebuttal) that 2D cooling is also made possible by multimode phase noise eating and the use of highly anisotropic particles relative to *Dania et. al.*. Some minor revision could at least partially resolve these issues.

6. (line 135) “...we split the transmitted cavity light into its eigenmodes...” Since this is a description of experimental method, the authors should describe *how* they split the polarizations. (Maybe I missed it?)

7. (line 186+) “To reduce the effect of phase noise, we measure it using a MZI whose output is fed back into an EOM. In this way...” As mentioned in my first review, phase noise is reduced by feedback, not by measurement. A native English speaker who reads this sentence will think it's the latter, which is incorrect. Since this is a key point, I recommend fixing the grammar.

8. (line 239) “we determine the SDEV of the rotational mode as ... corresponding to an effective temperature as low as $T = 28 \pm 2 \mu\text{K}$.” Two comments: (1) I believe that the authors mean “motion” (more precisely, librational amplitude), not “mode”; (2) if “as low as $T = 28 \mu\text{K}$ ” is meant as a cross-check of the sideband thermometry measurement, it begs the question whether 28 μK is consistent with $n = 0.21$ on line 213. Is it? Consider stating in the text.

The next few comments concern the conclusion, which is still difficult to understand as written.

9. (line 355) “We have demonstrated ground state cooling of both rotational DOFs individually...” Both rotational DOFs of what? Please state explicitly what was accomplished.

10. (line 359) “we align the nanorotor...” Which nanorotor? Please state explicitly.

11. (line 362) “Combined with the capability for fast and repeatable loading and cooling... these results provide...” It is unclear if this is a general statement, or if the authors are stating that “fast and repeatable loading...” is one of their accomplishments. Please be explicit.

12. (line 366) “With a mass around 4 GDa...” This sentence (and therefore paragraph) seems to come out of nowhere. The unit “GDa” is not previously used or defined in the text (it is implicitly defined later in the sentence by equating 10 MDa with the mass of two 20 nm silica spheres), nor have the authors stated in the conclusion or elsewhere in the text whether the spectrum of devices studied includes a 4 GDa rotor. Please be explicit to help guide the reader.

13. (line 381-382) “In addition, quantum nanorotors in the mass range of 0.1 – 1 MDa” Ditto. How does this relate to the present work? Please make the connection explicit.

14. (line 385) “The mass scale of 10 MDa...” Ditto.